

# Interannual Variability and Trends of Combustion Aerosol and Dust in Major Continental Outflows Revealed by MODIS Retrievals and CAM5 Simulations During 2003 – 2017

Hongbin Yu[1], Yang Yang[2], Hailong Wang[2], Qian Tan[3,4], Mian Chin[1], Robert C. Levy[1], Lorraine A. Remer[5], Steven J. Smith[2], Tianle Yuan[1,5], Yingxi Shi[1,5]

[1] Earth Sciences Division, NASA Goddard Space Flight Center, USA
[2] Pacific Northwest National Laboratory, USA
[3] Bay Area Environmental Research Institute, USA
[4] NASA Ames Research Center, USA
[5] Joint Center for Earth Science & Technology, University of Maryland at Baltimore County, USA

*Correspondence to*: Hongbin Yu (Hongbin.Yu@nasa.gov)

**Abstract.** Emissions and long-range transport of mineral dust and combustion-related aerosol from burning fossil fuels and biomass vary from year to year, driven by the evolution of the economy and changes in meteorological conditions and environmental regulations. This study offers both satellite and model perspectives of interannual variability and possible trend of combustion aerosol and dust in major continental outflow regions over the past 15 years (2003-2017). The decade-long record of aerosol optical depth (AOD, denoted as $\tau$), separately for combustion aerosol ($\tau_c$) and dust ($\tau_d$), over global oceans is derived from the Collection 6 aerosol products of the Moderate Resolution Imaging Spectroradiometer (MODIS) onboard both Terra and Aqua. These MODIS/Aqua datasets, complemented by aerosol source-tagged simulations using the Community Atmospheric Model Version 5 (CAM5), are then analyzed to understand the interannual variability and potential trend of $\tau_c$ and $\tau_d$ in the major continental outflows. Both MODIS and CAM5 consistently yield a similar decreasing trend of -0.017 to -0.020 decade$^{-1}$ for $\tau_c$ over the North Atlantic Ocean and the Mediterranean Sea that is attributable to reduced emissions from North America and Europe, respectively. On the contrary, both MODIS and CAM5 display an increasing trend of +0.017 to +0.036 decade$^{-1}$ for $\tau_c$ over the tropical Indian Ocean, the Bay of Bengal, and the Arabian Sea, which reflects the influence of increased anthropogenic emissions from South Asia and Middle East in the last two decades. Over the northwestern Pacific Ocean that is often affected by East Asian emissions of pollution and dust, the MODIS retrievals show a decreasing trend of -0.021 decade$^{-1}$ for $\tau_c$ and -0.012 decade$^{-1}$ for $\tau_d$, which is however not reproduced by the CAM5 model. In other outflow regions strongly influenced by biomass burning smoke or dust, both MODIS retrievals and CAM5 simulations show no statistically significant trends; and the MODIS observed interannual variability is usually larger than that of the CAM5 simulation.



## 1. Introduction

Mineral dust resulting from wind erosion in arid and semi-arid regions and combustion-related aerosol from burning fossil fuels and biomass are transported at intercontinental and hemispherical scales, and deposited into adjacent oceans in large amounts (Yu et al., 2013a; Uno et al., 2009). These aerosols exert important and far-reaching impacts on a variety of aspects of the environment, including air quality and human health (Prospero, 1999; Chin et al., 2007; Anenberg et al., 2014; Tao et al., 2016), radiation budget (Yu et al., 2006, 2012, 2013b; Song et al., 2018; Yang et al., 2017a, 2018a), cloud lifecycles and precipitation (Kaufman et al., 2005a; Creamean et al., 2013; Wang et al., 2013; Lu et al., 2018), terrestrial and marine ecosystems (Jickells et al., 2005; Yu et al., 2015a), and weather and climate (Yuan et al., 2016; Tao et al., 2018).

Emissions of combustion aerosol and its precursors associated with industrial activities have been changing in response to changes in population, energy structure, and environmental policies. Emissions of particles from biomass burning change with both atmospheric state and human practices. Dust emissions depend strongly on winds and synoptic system meteorology, as well as surface conditions. Meteorological conditions also determine where the particles go and how much is transported across oceans or deposited into oceans. Given large interannual variations in atmospheric circulations and changes in environmental regulations and other drivers on national to regional bases, it is expected that combustion aerosol and dust in the atmosphere may have experienced pronounced year-to-year variation and significant regional trends may have occurred. Such aerosol changes could have important implications for a variety of environmental issues as mentioned above. Currently it remains a great challenge to quantify the changes in aerosol sources, long-range transport, and environmental impacts.

Satellites are a suitable platform to observe aerosol interannual variability and trends because of their routine sampling over decadal and even multi-decadal time scales with extensive spatial coverage. For example, the Moderate Resolution Imaging Spectroradiometer (MODIS) sensor has been observing the global aerosol system from both Terra (since February 2000) and Aqua (since July 2002) satellites for more than 15 years (Levy et al., 2018). The MODIS aerosol data record could potentially be extended beyond its lifetime by bridging with retrievals applied to the Visible Infrared Imaging Radiometer Suite (VIIRS) onboard SNPP (launched in late 2011). VIIRS is sufficiently similar in design to MODIS that the MODIS aerosol retrieval algorithms have been adapted for VIIRS inputs, and continuity of the aerosol record is being evaluated (Levy et al., 2015). Several other sensors have also been detecting aerosols over the globe for at least a decade, including the Advanced Very High Resolution Radiometer (AVHRR) (Zhao et al., 2008), the Total Ozone Mapping Spectrometer (TOMS) (Torres et al., 2002), the Sea-Viewing Wide Field-of-view Sensor (SeaWiFS) (Hsu et al., 2012), the Multi-angle Imaging SpectroRadiometer (MISR) (Kahn et al., 2007), the Ozone Monitoring Instrument (OMI) (Torres et al., 2007), the Cloud Aerosol Lidar with Orthogonal Polarization (CALIOP) (Winker et al., 2013), and the Infrared Atmospheric Sounding Interferometer (IASI) (Capelle et al., 2018). These instruments have been acquiring long-term datasets of aerosol optical depth (AOD) and several particle properties (e.g., size, shape, and absorption). CALIOP also provides critical information for the aerosol vertical





distribution. These data records provide an opportunity for characterizing aerosol interannual variability and trends over a
decadal time scale, although various uncertainties in aerosol retrievals and sensor degradation could complicate trend detection
(Li et al., 2009).

In the last decade, a number of studies have analyzed satellite data to detect aerosol interannual variability and trends on
regional or global scales, mostly focusing on the total aerosol without distinguishing aerosol types from the data (Mishchenko
et al., 2007; Zhao et al., 2008; Zhang et al., 2010; Hsu et al., 2012; Chin et al., 2014; Jongeward et al., 2016; Alfaro-Contreras
et al., 2017; Zhang et al., 2017; de Leeuw et al., 2018; Chedin et al., 2018). Determining the types of aerosol responsible for
the detected AOD variability or trends remains elusive. A distinction between largely natural dust and largely man-made
combustion aerosol should provide greater insight into changes in aerosol sources and long-range transport than does total
aerosol. Dust and combustion aerosol are also different in their many properties and a variety of environmental impacts.
The overall objective of this study is to characterize the interannual variability and possible trend of combustion aerosol and
dust, separately, in major continental outflow regions over the last two decades. Specific science questions we address include:
How has industrial pollution changed on a regional basis in recent decades? How do episodic dust and biomass burning smoke
vary from year to year? Are there regional trends over the past 15 years? Why does the aerosol change? Is the AOD trend
consistent with that of aerosol emissions? We proceed to achieve the objective through a combined analysis of satellite
retrievals from the MODIS instrument and model simulations from the Community Atmosphere Model version 5 (CAM5). In
contrast to the previous work, this study examines combustion aerosol and dust separately from the MODIS retrievals. We
also use the CAM5 simulations with aerosol source-tagging capability to do source attribution and chemical characterization
for the MODIS-detected combustion AOD variability and trends in major outflow regions of continental aerosol. We focus on
the outflow regions over water bodies because the MODIS retrievals of AOD and particle properties are more accurate over
ocean than over land and thus we can better separate combustion aerosol from dust.

The rest of paper is organized as follows. Section 2 gives a brief description of the MODIS aerosol products and the CAM5
model setups. In Section 3, we first describe the derivation of dust and combustion aerosol components specific to the MODIS
Collection 6 (C6) data (3.1), which is an important update to previous studies based on the earlier MODIS data collections
(Kaufman et al., 2005b; Yu et al., 2009). This is followed by the seasonal and spatial characteristics of dust and combustion
AOD climatology based on both MODIS retrievals and CAM5 simulations (3.2). Then we analyze the interannual variability
and possible trend of $\tau_c$ and $\tau_d$ in major combustion aerosol and dust outflows from the MODIS/Aqua retrievals complemented
by the CAM5 simulations (3.3). Section 4 summarizes major conclusions from the combined MODIS and CAM5 analysis and
discuss major issues of relevance.

## 2.    Description of MODIS Retrievals and CAM5 Simulations of Aerosols

## 2.1. MODIS Aerosol Retrievals



The twin MODIS sensors have been flying on Terra (with equatorial overpassing time at about 10:30 am local time) since
February 2000 and on Aqua (with equatorial overpassing time at about 01:30 pm local time) since June 2002. The MODIS
instrument is a 36-channel radiometer covering wavelengths from deep blue to the thermal infrared. The instrument samples
a broad swath of about 2330 km at a spatial resolution of 500 m or 1 km (depending on channel) at nadir. As such the MODIS
sensors have been acquiring essential data for atmospheric and surface properties with nearly daily global coverage for a period
of more than 15 years. Three complementary aerosol retrieval algorithms, namely the dark-target (DT), the deep-blue (DB),
and the Multi-Angle Implementation of Atmospheric Correction (MAIAC) algorithms, have been developed, operated,
maintained, and updated at NASA Goddard Space Flight Center. The consistent retrieval algorithms have been applied to both
MODIS sensors during their lifetime. In this study, we use aerosol retrievals from the DT algorithm that was developed to
retrieve aerosol loading and properties over dark surfaces, including ocean/water *(Remer et al., 2005)* and vegetated land *(Levy
et al., 2007)*. Because of its wide spectral range and the simplicity of the dark ocean surface, the DT algorithm has the capability
of retrieving AOD with a high accuracy and information on particle size (in the form of Angstrom exponent, effective radius
or fine-mode fraction - FMF) (Remer et al., 2005; Levy et al., 2013).

In this study, we use the MODIS DT C6 over-ocean aerosol data records from both Terra and Aqua, i.e., the Level 3 daily
product gridded to 1°x1° (i.e., MOD08D3 and MYD08D3, respectively). In previous studies, we developed an approach of
distinguishing dust from combustion aerosol by using MODIS retrievals of total AOD ($\tau$) and FMF ($f$) (Kaufman et al., 2005b;
Yu et al., 2009). Both $\tau$ and $f$ refer to properties at 550 nm hereafter, except if specified otherwise. In this approach, $\tau$ and fine-
mode AOD ($f\tau$) are assumed to be composed of marine aerosol, dust, and combustion aerosol, i.e.,

$$\tau = \tau_m + \tau_d + \tau_c \qquad (1)$$

$$f\,\tau = f_m\,\tau_m + f_d\,\tau_d + f_c\,\tau_c \qquad (2)$$

where subscript m, d, and c refers to marine aerosol, dust, and combustion aerosol, respectively. The marine aerosol refers
collectively to the primary and secondary particles associated with wave-breaking sea-spay, including sea salt, marine organics,
and sulfate produced from dimethyl sulfide (DMS). The combustion aerosol, also referred to as "pollution" or "anthropogenic"
aerosol in the previous studies (Kaufman et al., 2005b; Yu et al., 2008, 2009), result mainly from burning of fossil fuels and
biomasses. It also includes contributions from volcanic activities. Based on equations (1) and (2), $\tau_d$ and $\tau_c$ can be calculated
from the MODIS retrieved $\tau$ and $f$, with appropriate parameterizations for $f_d$, $f_c$, $f_m$, and $\tau_m$ (see details in Kaufman et al., 2005b;
Yu et al., 2009). In earlier versions (e.g., C4 and C5), while $\tau_m$ was parameterized as a function of wind speed, $f_d$, $f_c$, and $f_m$
were determined from retrieved $f$ in selected regions and seasons where a specific aerosol type dominates (Kaufman et al.,
2005b; Yu et al., 2009). Given that the MODIS retrievals, in particular $f$, are sensitive to many details of the retrieval algorithm
and instrument calibration, the characteristic $f_d$, $f_c$, and $f_m$ should be considered as "dynamical". As discussed in Levy et al.
(2013, 2018), several aspects of the MODIS DT retrieval algorithm and the instrument calibration have evolved from C5 to



C6, which warrants a reassessment of the dust-combustion separation approach to assure the consistent use of the MODIS C6
product without possibly introducing additional errors (Yu et al., 2009). This is addressed in section 3.1.

## 2.2. CAM5 Simulations

To investigate interannual variability and trend of global aerosols, a 37-year simulation covering 1979–2015 has been carried
out with time-varying emissions of aerosol and precursors using the Community Atmosphere Model version 5 (CAM5-
MAM3). In the three-mode modal aerosol module (MAM3, Liu et al., 2012), aerosol species, including sulfate ($SO_4$), black
carbon (BC), primary organic matter (POM), second organic aerosol (SOA), mineral dust and sea salt, are predicted with
improved convective transport and wet scavenging schemes (Wang et al., 2013). At a 1.9° latitude by 2.5° longitude horizontal
grids and 30 vertical layers, model wind fields are nudged to the NASA Modern Era Retrospective-Analysis for Research and
Applications (MERRA) reanalysis (Rienecker et al., 2011) every 6-hours. This long-term historical simulation ended in 2015
because of the lack of availability of the MERRA reanalysis. Monthly anthropogenic (version 20160726) and open biomass
burning (version 20161213) emissions are taken from the Coupled Model Intercomparison Project Phase 6 (CMIP6) emission
database (Hoesly et al., 2018; van Marle et al., 2017). Note that the term biomass burning in this work refers to open burning,
such as forest fires and agricultural waste burning on fields. Biofuel combustion is categorized as anthropogenic emissions.
To understand the source attribution and chemical components of combustion aerosol in each of 13 continental outflow regions
(Figure 1a) in this study, we utilize the CAM5 aerosol source tagging capability (Wang et al., 2014; Yang et al., 2017b, 2017c,
2018a, 2018b). With this capability, BC, POM, sulfate (and precursor gases $SO_2$ and DMS) from a variety of sources are
explicitly tagged and tracked in the model. This tagging technique can offer insights into contributions by different
regions/sources, which satellite retrievals cannot provide. For this study, emissions of $SO_2$, BC, and POM are tagged with
respect to 14 source regions (Figure 1b) as defined in the Hemispheric Transport of Air Pollution Phase 2 (HTAP2), namely
North America (NAM), Central America (CAM), South America (SAM), Europe (EUR), North Africa (NAF), Southern Africa
(SAF), the Middle East (MDE), Southeast Asia (SEA), Central Asia (CAS), South Asia (SAS), East Asia (EAS),
Russia/Belarus/Ukraine (RBU, hereafter Russia), Pacific/Australia/New Zealand (PAN, hereafter the southern Pacific), and
rest of the world (ROW). The tagging also applies to two specific sources for sulfate: volcanic emissions (VOL), and DMS
chemistry (DMS). Note that secondary organic aerosol (SOA) is not included in the tagging in this study, although this is
calculated by the model. Figure 1c shows the linkage between the continental outflow regions and source regions as reflected
by the 2003-2015 climatology of CAM5 source attributions of combustion AOD in each of the outflow regions.

Figure 2 displays the global annual anthropogenic and biomass burning emissions of $SO_2$, BC, and POM from 2003 to 2015
that are stacked from the 14 source regions. There are several prominent changes of regional emissions. The $SO_2$ emissions
from North America and Europe have been consistently decreasing from 2003 to 2015, with an accumulated reduction of 67%
and 45% of their 2003 levels, respectively. On the contrary, the $SO_2$ emissions from South Asia have increased by 80%,
continuously over this time period. The $SO_2$ emissions from East Asia have taken a more complicated trajectory, increasing


from 2003 until 2007 and then fluctuating from year to year until a sharp decline occurred in 2015. Overall, the East Asia $SO_2$
emissions have decreased by 16% from 2003 to 2015. The different trajectories between East Asia and South Asia $SO_2$
emissions are generally consistent with satellite observations (Li et al., 2017). East Asia and South Africa are the two largest
emitters of BC and POM, with the former dominated by anthropogenic activities and the latter by biomass burning. In contrast
to the region's $SO_2$ emissions discussed earlier, BC and POM emissions in East Asia increased continuously by 40% and 27%,
respectively, in this dataset.  For both BC and POM emissions, large interannual variability is shown in major biomass burning
regions, such as South America, South Africa, Southeast Asia, and Russia.

## 3.    Results

### 3.1.  Characteristic FMFs for distinct aerosol types from MODIS C6 products

As discussed in 2.1, a caveat for the MODIS-based dust-combustion separation algorithm is that $f_d$, $f_c$, $f_m$ derived from the
MODIS/Terra C4 (Kaufman et al., 2005b) and C5 (Yu et al., 2009) products should not be used directly for the MODIS C6 on
Terra and Aqua. To maintain a self-consistent use of the MODIS C6 products, here we first reassess the characteristic FMF
for individual aerosol types (i.e., $f_d$, $f_c$, $f_m$) and then re-derive combustion and dust AOD by following the method in previous
studies *(Kaufman et al., 2005b; Yu et al., 2009)*. Figure 3 shows how the MODIS retrieved FMF changes with AOD, separately
for Terra (a) and Aqua (b), in three regions dominated respectively by outflows of combustion aerosol (North Atlantic Ocean,
in summer), dust (eastern North Atlantic Ocean just off the coast of North Africa, in summer), and marine aerosol (southern
Indian Ocean, full year). (These regions are defined the same as the previous studies to facilitate a comparison.) The data in
the southern Indian Ocean are stratified into two time periods when the region can be somewhat influenced by transported
smoke in one time period (May – October, denoted as "Marine1") and by transported dust in the other (November – April,
denoted as "Marine2").  For comparison, similar FMF-AOD relationships (Figure 3c) are derived from the Goddard Chemistry
Aerosol Radiation Transport (GOCART) model. In the GOCART model, sulfate, POM, and BC are counted as fine-mode
aerosol, whereas dust and sea-salt include both fine-mode and coarse-mode with a division at a radius of 1 μm. Generally,
MODIS retrievals and GOCART simulations consistently show similar FMF changes with respect to AOD.  In the combustion-
dominated outflow region, FMF increases with increasing AOD, which is consistent with the addition of fine-mode combustion
particles into the relatively coarser (smaller FMF) background marine aerosol. In the dust outflow regions, on the contrary,
FMF decreases with increasing AOD that is consistent with the addition of coarser dust particles (hence smaller f) on the
background marine aerosol. For the marine aerosol dominated southern Indian Ocean, FMF remains relatively constant with
increasing AOD for AOD up to about 0.06, for both "Marine1" and "Marine2" clusters. Then FMF increases (decreases) with
increasing AOD for the "Marine1" ("Marine2") cluster. This bifurcating pattern reflects the influence of long-range transported
smoke and dust in "Marine1" and "Marine2" cluster, respectively.


We derive from Figure 3 the characteristic $f_c$, $f_d$, and $f_m$ for individual aerosol types as follows. On one hand, $f_c$ for combustion
aerosol and $f_d$ for dust are taken as the average f for the largest AOD bin in the combustion aerosol and dust dominated region,



respectively. In the largest AOD bin, a contribution from marine background is at a minimum and is neglected. On the other hand, $f_m$ for marine aerosol is the average f in the smallest AOD bin that represents the marine background. Table 1 lists the

characteristic $f_c$, $f_d$, and $f_m$ derived from the MODIS C6 data (both Terra and Aqua) and the GOCART model. The three datasets consistently show that $f_c$ is substantially larger than $f_d$, with $f_m$ in between. However, some notable differences do exist among them. The MODIS-based characteristic $f_c$, $f_d$, and $f_m$ differs by 0.03, 0.05, and 0.07, respectively, between Terra and Aqua. These differences may have resulted from difference in instrument calibrations (Levy et al., 2018). The major difference between GOCART and MODIS exists for $f_m$, with the GOCART $f_m$ of 0.78 being much higher than the MODIS values of 0.55

for Terra and 0.48 for Aqua.

The characteristic $f_c$, $f_d$, $f_m$ derived from the MODIS/Terra C6 products in this study are also compared with those previously derived from the C4 (Kaufman et al., 2005b) and C5 (Yu et al., 2009), as shown in Table 2. Although $f_c$ remains nearly unchanged, notable but opposite changes have occurred for $f_d$ and $f_m$ in the course of evolving data collections. On one hand,

$f_d$ has decreased from 0.51 (C4) and 0.37 (C5) to 0.26 (C6). On the other hand, $f_m$ has increased from 0.32 (C4) and 0.45 (C5) to 0.55 (C6). These changes manifest the importance of the reassessment above to assure a consistent use of the MODIS products. Applying the characteristic FMF values derived from the earlier versions of data (C5 or C4) to the MODIS C6 products would introduce large uncertainty to the calculation of $\tau_d$ and $\tau_c$. Similarly, caution should be exercised when applying the MODIS-based characteristic FMF values to the VIIRS over-ocean aerosol products, despite their similarity in instrument

design and the dark target aerosol retrieval algorithm (Levy et al., 2015). It is strongly recommended that similar assessment of the characteristic FMFs be carried out with the VIIRS retrievals.

### 3.2. Climatology of combustion aerosol and dust from MODIS and CAM5

We apply the C6-based characteristic FMF values to derive $\tau_c$ and $\tau_d$ from the MODIS/Terra and MODIS/Aqua C6 products

by following a similar method described in Yu et al. (2009). $\tau_m$ is parameterized as a function of daily 10 m wind speed from the Modern-Era Retrospective analysis for Research and Applications, Version 2 (MERRA-2) (Geralo et al., 2017) following the average of two relationships from Kiliyanpilakkil and Meskhidze (2011) and Mulcahy et al. (2008). Figure 4 shows the 2003-2015 MODIS/Aqua climatology of annual mean total AOD and its partition into $\tau_c$, $\tau_d$, and $\tau_m$ over ocean (left panels). The figure shows that the spatial patterns of combustion and dust AOD over ocean derived from the MODIS retrievals are

consistent with their largely well-known upwind continental sources. Major outflow regions with the elevated combustion AOD include: off the coast of the Asian continent over the North Pacific, off the coast of the eastern US over the North Atlantic, off the coast of the Indian subcontinent over the tropical Indian Ocean and the Bay of Bengal, in the Gulf of Mexico and eastern tropical Pacific Ocean, and over the tropical Atlantic Ocean downwind of the African biomass burning region. Major dust outflow regions include: the tropical Atlantic Ocean and the Caribbean Basin, the North Pacific Ocean, and the

Arabian Sea/tropical Indian Ocean. Although dust has been observed over the southeastern Atlantic Ocean, the MODIS-



derived dust AOD is likely overestimated. The year-round presence of stratocumulus clouds in the region poses a major challenge for the aerosol retrievals, probably yielding a high AOD bias, low FMF bias, and hence a high bias of dust AOD. Over the tropical Pacific Ocean, a long belt of somewhat elevated dust AOD may also indicate an artifact due to possible cloud contamination. The seasonal variations of $\tau_c$ and $\tau_d$ are shown in Figure S1 of the Supplementary Online Material (SOM).


The CAM5 simulations of $\tau$ and its components ($\tau_c$, $\tau_d$, and $\tau_m$) over both land and ocean are shown in the right panels of Figure 4. Here for the purpose of comparison with the MODIS data, we derive the CAM5 marine AOD as a sum of AOD of sea-salt and DMS-generated $SO_4$, and the combustion AOD as a sum of AOD of $SO_4$ (excluding those generated from DMS chemistry), BC, POM, and SOA. The CAM5 model captures the major outflows of combustion aerosol well. On the global

ocean average, the CAM5 $\tau_c$ (0.033) is lower than the MODIS retrieval of 0.040 by about 17%. However, the CAM5 model substantially underestimates the dust outflows into North Pacific Ocean and the tropical Atlantic Ocean, although it reproduces the MODIS-derived dust outflow into the Arabian Sea and the tropical Indian Ocean. On the basis of global ocean average, the MODIS-derived $\tau_d$ of 0.048 is about a factor of 5 higher than the 0.009 from the CAM5 model. The MODIS-derived global mean $\tau_m$ is 0.066 at 550 nm, which agrees well with the climatology based on AERONET observations (Kaufman et al., 2001;

Smirnov et al., 2002). Although the CAM5 model agrees with the global mean of MODIS-derived marine AOD within 8%, the model is higher at mid-high latitudes but lower in tropical regions than the MODIS-derived marine AOD. In summary, substantial disparity of dust AOD exists between the MODIS retrieval and CAM5 simulations. Although the cloud contamination issues discussed earlier may have biased the MODIS-derived dust AOD higher, this alone is unlikely to adequately explain the factor of 5 difference between MODIS retrievals and CAM5 simulations. We will further discuss this

issue by applying insight gained from a regional analysis.

The partitioning of AOD is further examined on a regional basis, focusing on 13 major continental outflow regions and two pristine regions as illustrated in Figure 1a. Figure 5 shows the regional partition of total AOD into marine aerosol, combustion aerosol, and dust on an annual mean basis from both MODIS/Aqua retrievals and CAM5 simulations. Both absolute value and

fractional contribution of AOD are shown. Seasonal variability of regional aerosol components based on the MODIS retrievals is shown in Figure S2 of SOM. The MODIS retrievals show that marine AOD falls into a range of 0.053-0.067 across all the regions. In the pristine regions (STP and STI), $\tau_m$ accounts for about 60% of total AOD. The remaining is partitioned into $\tau_c$ of 0.015 and $\tau_d$ of 0.024 on average. For comparison, the CAM5 simulations suggest that marine AOD accounts for about 90% of total AOD in these pristine regions, with the remaining 10% contributed largely by combustion aerosol. Although the

MODIS-retrieved $\tau_c$ and $\tau_d$ levels in these pristine regions do represent the smallest continental influences among all the regions, they may be subject to uncertainty or bias. For example, the MODIS aerosol retrievals are inevitably contaminated by imperfect cloud screening, possibly resulting in a high AOD bias and low FMF bias. The parameterization of $\tau_m$ could also introduce uncertainty to the derived $\tau_c$ and $\tau_d$. Both the issues would affect $\tau_d$ more than $\tau_c$. It is however formidable to





rigorously assess the bias without acquiring independent measurements of dust optical depth with high accuracy. Nevertheless,
if we assume that these remote regions are perfectly pristine without any influence from combustion aerosol and dust, then the
0.015 and 0.024 could represent an upper bound of potential bias associated with the MODIS-derived $\tau_c$ and $\tau_d$, respectively.
If it is further assumed that similar magnitude of the bias exists over the global ocean, this yields a bracket for the MODIS-
based global-ocean average of 0.025 - 0.040 for $\tau_c$ and 0.024 - 0.048 for $\tau_d$. For comparison, $\tau_c$ derived from the earlier versions
of the MODIS product is 0.033 (Kaufman et al., 2005c) and 0.035 (Yu et al., 2009) for C4 and C5, respectively. The CAM5
$\tau_c$ of 0.033 also falls into the range of the adjusted MODIS retrievals. On the other hand, the $\tau_d$ of 0.009 from the CAM5 model
still accounts for no more than 38% of the MODIS retrieval. This may suggest a great deal of effort is needed to improve the
model simulations of dust.

In comparison to the remote regions, the percentage of $\tau_m$ contribution from the MODIS retrieval is decreased to 13 – 47 % in
the major outflow regions, with the magnitude depending on the influence of continental aerosol sources. The $\tau_m$ contribution
is more than 40% in the northeastern Pacific Ocean (NEP) and the northern Atlantic Ocean (NAT). The $\tau_m$ contribution drops
below 20% in the Gulf of Guinea (GOG), the Arabian Sea (ARB), northern Indian Ocean and the Bay of Bengal (IND), and
the tropical Atlantic Ocean (TAT), because of strong influences of desert dust and combustion-related aerosol from the upwind
continent. The MODIS retrievals in Figure 5 further indicate relative contributions of $\tau_c$ and $\tau_d$ in the outflow regions. The
three most dust-dominated regions are TAT, GOG, ARB where $\tau_d > 0.2$ and accounts for more than 50% of $\tau$. The three highest
regional $\tau_c$ (> 0.1) occurs in IND, GOG, and the northwestern Pacific Ocean (NWP), which respectively accounts for about
44%, 28%, and 45% of $\tau$. The lower percentage of $\tau_c$ in GOG results from the co-existence of high $\tau_d$ of 0.237 in the region.
Several regions have a lower $\tau_c$ that however accounts for a higher percentage of $\tau$ due to small share of dust, including the
tropical eastern Pacific (TEP), the southeastern Atlantic Ocean (SAT), the Mediterranean Sea (MED), the southeastern Asia
(SEA), and the tropical western Pacific Ocean (TWP). As discussed earlier, the contribution of $\tau_c$ in SAT may have been
underestimated, because the persistence of high cloudiness in the region likely has resulted in the high bias to $\tau$ and $\tau_d$. In
comparison to the MODIS retrievals, the CAM5 simulations generally yield much smaller dust fractions. One exception is the
MED region where CAM5 dust fraction is the largest among all the regions.

Although the MODIS/Terra retrievals (not shown) bear great resemblance to the patterns revealed by the MODIS/Aqua
retrievals (Figure 4), the two differ somewhat in the magnitude of AOD. Previous studies have found an increasing trend of
total AOD over global ocean from analyzing the MODIS/Terra retrievals, but no trend from the MODIS/Aqua retrievals
(Zhang et al., 2017; Levy et al., 2018). It has been argued that this MODIS/Terra trend of global ocean mean AOD is spurious,
which could have been resulted from imperfect calibrations being implemented to correct radiances observed by the aging
Terra platform, different sampling due to changing clouds and viewing angles from late morning to early afternoon, among
other issues (Levy et al., 2018). Here we further explore this issue by examining if this spurious trend affects combustion and



dust AOD differently. Figure 6 compares the time series of MODIS/Terra and MODIS/Aqua monthly retrievals averaged over global oceans (60°S-60°N) for total, fine-mode, combustion, and dust AOD. Although the twin MODIS sensors observe the consistent month-to-month variations for both total and component AODs, clear offsets exist between Terra and Aqua datasets,

with the MODIS/Terra values being higher than the MODIS/Aqua values in general. The offsets have also become larger since 2007 (particularly after 2012) for total, fine-mode, and combustion AOD. Furthermore, an increasing trend is clearly shown in the MODIS/Terra retrievals of total, fine-mode, and combustion AOD, but not evident in the MODIS/Aqua retrievals. Although the dust AOD, in particular its minimum value of annual cycle, is lower in Aqua than in Terra, there is no clear trend in both MODIS/Terra and MODIS/Aqua retrievals. The increasing trends in the MODIS/Terra retrievals but lack of any in the

MODIS/Aqua retrievals do not represent realistic late morning and early afternoon differences of aerosol, in particular on the basis of global ocean average (Levy et al., 2018). As a sanity check, we further examine AODs in the two remote regions, i.e., STP and STI (as defined in Figure 1a), where the influence by continental outflows of aerosol is minimal (as clearly shown in Figures 4 and 5). A similar strategy has been applied in the literature (e.g., Zhang and Reid, 2010; Zhang et al., 2017) and Alfaro-Contreras et al. (2017) adjusted the MODIS/Terra AOD trends by subtracting the detected trend in the remote-ocean

regions. As shown in Figure 7, the MODIS/Aqua retrievals are consistent with the anticipation that the combustion AOD in such relatively pristine regions has had no consistent trend over the last two decades. On the contrary, the MODIS/Terra retrievals of $\tau_c$ show an increasing trend of +0.005 decade$^{-1}$ (p < 0.001) in both regions. For comparison, Zhang et al. (2017) detected an increasing trend of +0.006 decade$^{-1}$ for total AOD in the remote-ocean regions. Based on the above analysis, we will only use the MODIS/Aqua retrievals to examine interannual variability and possible trend of combustion and dust AOD

on a regional basis.

### 3.3. Interannual variability and trends of combustion aerosol and dust in major continental outflows

In this section, we use the MODIS/Aqua retrievals over the last 15 years (2003-2017) to study regional AOD interannual variability and trend, separately for combustion aerosol and dust, focusing on the 13 major continental outflow regions defined

in Figure 1a. The significance of a trend is checked with the student t-test at two levels (p < 0.01 or p < 0.05). The MODIS retrievals are compared with the CAM5 simulations over the 2003-2015 period. The CAM5 source-tagging simulations are used to facilitate the interpretation of the MODIS-observed $\tau_c$ changes in terms of major source regions and chemical composition. Although the same analysis has been performed for both $\tau_c$ and $\tau_d$ in all the outflow regions, we only discuss results with salient features in the following and place remaining results in SOM.


### 3.3.1. Declining trends of $\tau_c$ in North American and European outflows

Industrial and other anthropogenic emissions from North America (NAM) and Europe (EUR) have been declining steadily over the past several decades (see Figure 2). Several studies have also consistently observed the decreasing trend of total AOD. Hsu et al. (2012) derived a decreasing trend of -0.028 and -0.027 decade$^{-1}$ for $\tau$ over Eastern U.S. and Europe, respectively, by



using the SeaWiFS data over 1998-2010. In the east coast of North America, the MODIS/Aqua retrievals show a decreasing trend of -0.028 ~ -0.030 decade$^{-1}$ (Alfaro-Contreras et al., 2017; Jongeward et al., 2016), whereas both the MODIS/Terra and MISR data (after correcting the spurious increasing trend detected from the remote-ocean regions) yield a decreasing trend of -0.022 decade$^{-1}$ (Alfaro-Contreras et al., 2017). Over the Mediterranean Sea, Alfaro-Contreras et al. (2017) detected a decreasing trend of -0.025 (MODIS/Aqua and MISR) and -0.020 (MODIS/Terra) decade$^{-1}$. Figure 8 shows the MODIS/Aqua

retrievals and the CAM5 simulations of the climatological seasonal variations of $\tau_c$ (top panels), and the time series of annual-mean $\tau_c$ (middle panels) in the North Atlantic Ocean (NAT, left panels) and Mediterranean Sea (MED, right panels) regions. In these panels, stacked bars are included to show the CAM5 simulated relative contributions by sulfate, POM, BC, and SOA to the total AOD. The source contributions based on the CAM5 tagged-source simulations are shown in the bottom panels. In the NAT region, the peak influence of MODIS-retrieved $\tau_c$ occurs in July, which is about a factor of 4 larger than that in winter

months. The CAM5 model well captures the annual cycle of $\tau_c$ with similar amplitude, although the modeled $\tau_c$ is generally higher than the MODIS retrieval in winter and fall months. The CAM5 model also reveals that the summer $\tau_c$ peak results mainly from increased sulfate and SOA, presumably due to more active photochemistry and higher relative humidity in summer. In the MED region, the peak occurs in August for the MODIS, but September for the CAM5. In winter and fall months, the CAM5 $\tau_c$ is higher than the MODIS retrieval, yielding a smaller amplitude of the CAM5 annual cycle. Similar to

NAT region, the summer $\tau_c$ peak in MED region also results from increased sulfate and SOA. In both NAT and MED regions, the MODIS and CAM5 show consistently the decreasing trends of annual $\tau_c$ from 2003 onward, at a level of -0.017 ~ -0.020 decade$^{-1}$(p < 0.01), which is somewhat smaller than those for total optical depth $\tau$ using similar satellite data (Alfaro-Contreras et al., 2017; Jongeward et al., 2016). The CAM5 source-attribution simulations suggest that about 95% of $\tau_c$ variability can be explained by man-made sulfate from a single source region, i.e., North America for NAT and Europe for MED. In the MED

outflow region, the volcanic emission (VOL) constitutes the second largest contribution, which however has no clear trend over the period.

The decreasing trends of $\tau_c$ in the NAT and MED outflow regions also depend on season, as shown in Table 3. In the NAT region, the MODIS/Aqua retrievals show a decreasing trend of -0.040 decade$^{-1}$ (p < 0.01) and -0.021 decade$^{-1}$ (p < 0.01) in

JJA and MAM, respectively, which is much greater and more statistically significant than that in DJF (-0.005 decade$^{-1}$) and SON (-0.010 decade$^{-1}$). On the other hand, the CAM5 simulations show statistically significant trend (p < 0.01) in all season, however the decreasing trends of -0.023 and -0.033 decade$^{-1}$ in MAM and JJA are a factor of 2-3 stronger than the decreasing trends of -0.011 and -0.012 decade$^{-1}$ in DJF and SON. The finding of the stronger trend in JJA and MAM than in DJF and SON is consistent with that for total AOD from Jongeward et al. (2016). In the MED outflow region, the MODIS retrievals

show statistically significant trends of -0.027 decade$^{-1}$ (p < 0.01) and -0.025 decade$^{-1}$ (p < 0.01) in MAM and JJA, but -0.020 decade$^{-1}$ (p < 0.05) in SON. There is no significant trend in DJF. In this region, the CAM5 simulations are consistent with the





MODIS retrievals in terms of seasonal variability but with a smaller magnitude of the decreasing trend (e.g., -0.015 to -0.022 decade$^{-1}$ or 20-30% lower).

### 3.3.2. Increasing trends of $\tau_c$ in the outflows of South Asia and Middle East

On the contrary to North America and Europe where combustion emissions have been decreasing over the past decades, combustion emissions in South Asia (SAS) and Middle East (MDE) have increased relatively steadily (see Figure 2). Li et al. (2017) show with satellite data that India may have surpassed China in recent years by becoming the largest emitter of anthropogenic sulfur dioxide. Proestakis et al. (2018) detected from analyzing the CALIOP data an increasing trend of + 0.033 decade$^{-1}$ for total AOD $\tau$ in South Asia from 2007-2015. The SeaWiFS data suggest a strong increasing trend of +0.092 decade$^{-1}$ in the Arabian Peninsula and + 0.063 decade$^{-1}$ in northern India for total AOD during 1998-2010 (Hsu et al., 2012). We show in Figure 9 that the increase of combustion emissions in South Asia and Middle East has caused an increasing $\tau_c$ trend over the tropical Indian Ocean and the Bay of Bengal (IND) and the Arabian Sea (ARB). In the IND region, both the MODIS retrievals and the CAM5 simulations show higher $\tau_c$ during October-March months. Although the CAM5 $\tau_c$ is higher than the MODIS $\tau_c$ by about 0.035 or 35% in 2003, the difference has reduced since then, becoming negligible in 2013-2015. Thus, the MODIS retrievals yield a much larger increasing trend of + 0.036 decade$^{-1}$ (p < 0.01) for $\tau_c$, compared to the CAM5-simulated trend of + 0.020 decade$^{-1}$ (p < 0.05). For comparison, Alfaro-Contreras et al., (2017) derived a total AOD trend of + 0.031 (MODIS/Aqua), + 0.050 (MODIS/Terra), + 0.022 (MISR) $\tau$ decade$^{-1}$ over the Bay of Bengal. The CAM5 source attributions show that although South Asia sources are the largest contributor to the IND $\tau_c$ particularly in recent years, sources from other regions such as East Asia, Southeast Asia, and Middle East also make non-negligible contributions. These four source regions combined (including the SOA formation) explain 98.5% of $\tau_c$ variability.

In the ARB region, the MODIS retrievals show a peak $\tau_c$ in October-December, while the CAM5 simulations show an additional second peak in July-August. The seasonal variation of combustion aerosol is mainly driven by sulfate, as suggested by the CAM5 simulations. The CAM5 $\tau_c$ is consistently higher than the MODIS retrieval throughout the year, and on annual mean basis $\tau_c$ is higher by about +0.043 or 48%. Despite this difference in magnitude of $\tau_c$, an increasing trend of similar magnitude of +0.017 decade$^{-1}$ is derived from both the MODIS retrievals (p < 0.01) and the CAM5 simulations (p < 0.05). The CAM5 source-attribution simulations further identify that the increasing trend in the ARB regions is largely driven by the increase of emission from both South Asia and Middle East. These two regions combined explain 94.4% of $\tau_c$ variability. Our derived increasing $\tau_c$ trend of +0.017 decade$^{-1}$ is a factor of 2-3 smaller than the +0.037 to +0.051 decade$^{-1}$ for total AOD derived from previous work with MODIS (Terra and Aqua) and MISR retrievals (Alfaro-Contreras et al., 2017).

The trends in both regions show strong seasonal variations, as shown in Table 4. In the IND outflow region, both the MODIS retrievals and the CAM5 simulations suggest much stronger trends with higher statistical significance in the pre-monsoon





seasons (MAM and DJF) than during SON. In the ARB region, on the other hand, the MODIS/Aqua retrievals show a stronger

increasing trend of + 0.031 decade$^{-1}$ in SON than the + 0.023 ∼ 0.025 decade$^{-1}$ in DJF and MAM for $\tau_c$. On the contrary, the

CAM5 results show an increasing trend of + 0.031 decade$^{-1}$ ($p < 0.05$) and + 0.026 decade$^{-1}$ ($p < 0.01$) in MAM and DJF

respectively, but no significant trend in SON and JJA.

### 3.3.3. Non-monotonic trends of $\tau_c$ in East Asian outflows

East Asia is the largest emitter of combustion aerosol and precursors in the world, as shown in Figure 2. As discussed earlier,

however, the regional emissions have followed a complicated trajectory in recent decades.  de Leeuw et al. (2018) analyzed

satellite AOD data over mainland China from 1995-2015 and found a distinct increase of AOD from 1995 to 2000, but a

decreasing AOD since about 2011. Proestakis et al. (2018) found that total AOD over East China has decreased from 2007 to

2015 at a rate of -0.050 decade$^{-1}$ based on the CALIOP data.  Observations from a large environmental monitoring network

since 2013 have shown a general 30-50% decrease of annual mean PM$_{2.5}$ across China over the 2013-2018 period, which is

consistent with concurrent observations of SO$_2$ and CO (Zhai et al., 2019). While the decline PM$_{2.5}$ trend is mainly attributable

to drastic controls of coal combustions, meteorology makes secondary but still significant contribution to the decline trend of

PM$_{2.5}$ (Chen et al., 2018, 2019; Cheng et al., 2019; Zhai et al., 2019). How the outflows of East Asia emissions influence the

northwestern Pacific Ocean (NWP) and tropical western Pacific Ocean (TWP) is shown in **Figure 10** from both the MODIS

retrievals and CAM5 simulations. It reveals that the MODIS observed annual cycle in the NWP region has much higher $\tau_c$ in

March-July than in other months. The MODIS $\tau_c$ is higher than the CAM5 modeled $\tau_c$ by 35-53% during March-July, but the

two datasets agree within 4-22% in the other months. This results in a much weaker seasonal variation of $\tau_c$ in CAM5 than in

MODIS. Anderson et al. (2005) compared MODIS FMF with in-situ measurements of small-mode fraction during the 2001

ACE-Asia campaign and found that the MODIS FMF is biased high by about 0.2. Although about 20% of this high FMF bias

for that specific year could be due to anomalous behavior in the MODIS/Terra's "side-B" electronics, the rest would be

attributed to the assumption of spherical shape for dust in the MODIS retrieval (Anderson et al., 2005).  It is thus possibly that

the large MODIS-CAM5 difference during the March-July period could at least be partially attributed to the spherical dust

assumption in the MODIS algorithm.  Over the period of 2003-2017, the MODIS retrievals show a decreasing trend of -0.021

decade$^{-1}$ ($p < 0.05$) in NWP. The MODIS retrievals also appear to show a stronger decreasing trend in the latter part of this

period starting 2007. Our result is generally consistent with findings from other studies, in which the trajectory of AOD

changed from an increasing to a decreasing at certain pivot point. For example, Zhang et al. (2017) analyzed the MODIS data

over the northwestern Pacific Ocean and found that AOD has undergone a shift from an increasing trend over 2000-2007 to a

decreasing trend over 2007 - 2015. Alfaro-Contreras et al. (2017) analyzed several satellite retrievals of AOD for total aerosol

and found no consistent trend in coastal China (-0.035, +0.001, -0.010 decade$^{-1}$ for MODIS/Aqua, MODIS/Terra, and MISR,

respectively).  Sogacheva et al. [2018] combined ATSR and MODIS AOD data and found that AOD across China increased

significantly from 1995 to 2006 but decreased gradually between 2011 and 2017, which reflects the increased emissions due



to rapid economic development and the decreased emissions due to effective emission-control regulations. For comparison, the CAM5 simulations show no statistically significant trend. The CAM5 simulations also suggest that 95% of $\tau_c$ variability

can be explained by a single source region of East Asia.

In the TWP region, although the MODIS retrievals and CAM5 results agree well from May to December, they differ in January to April, the months of peak $\tau_c$. The MODIS retrievals show peaks in March-April, whereas the CAM5 simulation suggests peaks during January-March.  Over the period of 2003-2017, the MODIS retrievals show a decreasing trend of -0.023 decade$^{-1}$

($p < 0.01$) in TWP. Note that the MODIS $\tau_c$ in 2005 was higher by 31-92% than that in any other year. Previous studies (Yuan et al., 2011; Yu et al., 2008) have traced the 2005 high aerosol loading to volcanic activity east of the Philippines. Clearly, the CAM5 model underestimates this 2005 $\tau_c$ peak by 28%, although the model produces the magnitude of MODIS $\tau_c$ in the other years well. This may suggest that the CAM5 model may have underestimated the 2005 volcanic emissions in this region. Compared to the MODIS retrievals, the CAM5 simulations show no statistically significant trend of combustion AOD. Similar

to the NWP region, 95% of $\tau_c$ variability can be explained by emissions from East Asia.  Boreddy et al. (2018) examine the 2001-2012 data record of surface carbonaceous aerosols measured in Chichijima, a remote island (27°04' N, 142°13' E) just near the northeastern corner of our TWP box. Their data show that carbonaceous aerosol has maxima in winter to spring and minima in summer, which is consistent with the CAM5 model simulation. They also detected statistically significant increasing trends of OC and water-soluble OC due probably to the enhanced photochemical oxidation of biomass burning and biogenic

VOCs during the long-range transport, but a decreasing trend of elemental carbon (EC) probably associated with the decrease of fossil-fuel sources. For comparison, the CAM5 simulations over the same period shows weakly increasing but not statistically significant trends for BC, POM, and SOA. The above analyses suggest that the model would need to improve its simulations of combustion aerosol trends in this region.

**3.3.4. Interannual variability in smoke or dust dominated outflows**

In comparison to industrial emissions, biomass burning emissions often show large fluctuations from year to year but without a steady trend, because of their strong dependence on meteorological conditions. Figure 11 shows results from our analysis of the MODIS retrievals and CAM5 simulations in three outflow regions where biomass burning smoke contributes significantly to $\tau_c$, namely the Gulf of Guinea (GOG), the southeastern Atlantic Ocean (SAT), and southeastern Asia (SEA).  In all three

regions, the MODIS retrievals and CAM5 simulations show interannual variations but no trends.  The CAM5 simulations suggest that emissions from southern Africa can explain more than 94% and 97% of $\tau_c$ variability in the GOG and SAT outflow region, respectively. Although the CAM5 $\tau_c$ is on average lower than the MODIS retrieval by only 0.008 or 9% in the SAT outflow region, the CAM5 $\tau_c$ is 0.035 or 28% higher than the MODIS retrieval in the GOG outflow region. It is important to note that GOG and SAT are affected by biomass burning from the northern and southern equatorial Africa, respectively.

However, the northern and southern equatorial Africa has been aggregated together as a single broad source region (SAF) in





this study. It is possible that the biomass burning emission in the southern equatorial Africa is reasonably captured by CAM5, whereas the emission in the northern equatorial Africa may be overestimated by the model. On the other hand, the smaller MODIS $\tau_c$ in GOG might also allude to another possibility that our MODIS-based dust-combustion separation algorithm may have attributed part of combustion aerosol to dust [Yu et al., 2009].


In the SEA outflow region (Figure 11, bottom panels), the MODIS retrievals show very large interannual variability, with elevated $\tau_c$ values corresponding to El Nino years (e.g., 2004, 2006, 2009, and 2015) when drought conditions associated with El Nino driving up biomass burning events in Indonesia (Field et al., 2016; Pan et al., 2018). Year 2015 and 2006 had higher $\tau_c$ than year 2009 and 2004, which cannot be explained by the El Nino index alone, but instead is believed to result from

interactions of El Nino and Indian Ocean Dipole (IOD) that determine atmospheric circulations and rainfall (Pan et al., 2018). Specifically, 2015 and 2006 were years that IOD was in phase with El Nino (so called Eastern Pacific-type of El Nino), whereas 2009 and 2004 were years that IOD was out of phase or weakly in phase with El Nino (so called Central Pacific-type of El Nino) (Pan et al., 2018). In comparison, the CAM5 simulations are on average 42% lower than the MODIS retrievals and show much smaller interannual variability. Carbonaceous aerosol (POM, BC, and SOA) accounts for about half of $\tau_c$ in the region.

The source attribution analysis of the CAM5 simulations shows that sources from Southeast Asia and SOA combined explain about 88% of the $\tau_c$ variability. The substantially lower magnitude and smaller interannual variability of $\tau_c$ by CAM5 suggest that the fire emissions in Southeast Asia are likely to be underestimated in the model. In particular, the CAM5 POM emission from Southeast Asia in 2015 was even 25% lower than that in 2014, which is in stark contrast to the MODIS observations. It is also important to note that the POM/SOA ratio in the southeastern Asia is substantially smaller than that in the Gulf of

Guinea and southeastern Atlantic Ocean.

Similar to the biomass burning smoke outflow regions discussed above, $\tau_d$ in six major dust outflow regions also displays large interannual variability, but generally no clear trend, based on both the MODIS retrievals and CAM5 simulations, as shown in Figure 12. One exception is the northwestern Pacific (NWP) outflow region influenced mainly by Asian dust emissions where

MODIS $\tau_d$ shows a decreasing trend of -0.012 decade$^{-1}$ or -1.5% year$^{-1}$($p < 0.05$). Further analysis shows that this decreasing trend of annual $\tau_d$ results from a decreasing trend of -0.034 decade$^{-1}$ or -2.6% year$^{-1}$ ($p < 0.05$) in MAM but negligible trends in the other seasons. These decreasing trends of dust are smaller in magnitude than that detected from the Asian Dust Network (AD-Net) lidar observations over Japan, i.e., -4.3% year$^{-1}$ and 2.5% year$^{-1}$ for MAM and annual mean, respectively (Shimizu et al., 2017). Our results are also consistent with trends of dust emissions over East Asia and China in particular as documented

in literature. For example, Song et al. (2016) showed that the spring dust storm frequency in arid and semiarid regions of China has decreased by 15.45 storms per year on average over the period of 1982 to 2007. Fan et al. (2014) showed that the decrease of springtime dust storms in Inner Mongolia, northern China from 1982-2008 was correlated with the advanced vegetation growth, with one-day earlier green-up data corresponding to a 3% decrease of spring dust storms. An et al. [2018] analyzed



the frequency of sand and dust storms observed from ground stations in East Asia during 2007-2016 and found a decreasing trend that is associated with the increase of vegetation coverage and the weakening of the polar vortex. Proestakis et al. (2018) detected from analyzing the CALIOP data a stronger decreasing trend of -0.021 decade$^{-1}$ for annual average $\tau_d$ during the period of 2007-2015 in East China (a domain including both land and adjacent eastern China Sea). While Proestakis et al. (2018) also found a decreasing trend of -0.015 decade$^{-1}$ over South Asia (a domain including both India subcontinent, the Bay of Bengal, and northern Indian Ocean), the MODIS retrievals in our study show no statistically significant trend in the IND outflow region. Finally, although CAM5 $\tau_d$ in the MED outflow region is 36% higher than the MODIS retrieval, it is a factor 2-3 smaller than the MODIS retrieval in other dust outflow regions. This low bias of CAM5 $\tau_d$ in the regional analysis is expected from the global plots of Figure 4.

## 4. Conclusions and discussion

We have reassessed the MODIS FMF-based algorithm that partitions total AOD over ocean into combustion aerosol, dust, and marine aerosol by using the Collection 6 data from both Terra and Aqua. By using the derived C6-specific characteristic FMF for the individual aerosol types, we produced the MODIS/Terra and MODIS/Aqua retrievals of over-ocean $\tau_c$ and $\tau_d$ over the period of 15+ years. The MODIS retrievals were compared with the CAM5 simulations. We then analyzed the MODIS/Aqua retrievals from 2003 to 2017 to examine the interannual variability and possible trend of $\tau_c$ and $\tau_d$ over 13 major continental outflow regions around the globe. This MODIS-based analysis was complemented by the CAM5 source-tagging simulations from 2003 to 2015 to identify the source attributions. Major results derived from this study include:

1. The characteristic FMF for combustion aerosol, dust, and marine aerosol is derived, respectively, to be 0.89, 0.31, and 0.48 from the MODIS/Aqua C6 product. Corresponding values derived from the MODIS/Terra C6 product are 0.92, 0.26, and 0.55, respectively. Significant changes in the characteristic FMF values for dust and marine aerosol have occurred throughout the evolution from C4 and C5 to the latest C6. Differences in instrument calibrations and changes in the details of the aerosol retrievals may have created these differences between Aqua and Terra, and between different data collections. The bottom line is that the MODIS data in each collection should be used in a consistent way to avoid introducing unwanted errors. Applying the MODIS-based characteristic FMF directly to VIIRS observations is also not encouraged.

2. The CAM5 combustion AOD $\tau_c$ simulations show a close agreement with the MODIS retrievals over ocean, with the global ocean average being ~ 17% lower than the MODIS $\tau_c$. However, the CAM5 simulations of dust AOD $\tau_d$ over ocean are nearly a factor of 3 smaller than the MODIS retrievals even when possible cloud contamination in the MODIS retrievals are empirically accounted for.

3. In contrast to the MODIS/Aqua retrievals, the MODIS/Terra retrievals show a statistically significant, increasing trend of combustion AOD in the remote-ocean regions and global ocean average. This trend is considered to be spurious, due possibly to the imperfect instrument calibrations of the MODIS/Terra instrument and other issues,





as presented and discussed in *Levy et al. [2018]*. It is thus recommended that only the MODIS/Aqua retrievals only be used to examine regional AOD trends.

4. The MODIS retrievals and CAM5 simulations consistently yield a decreasing trend of -0.017 to -0.020 decade$^{-1}$ for the combustion AOD ($p < 0.01$) over the North Atlantic Ocean (NAT) and the Mediterranean Sea (MED) that is respectively influenced by the reduction of combustion-related emissions from North America and Europe. In these regions, MODIS and CAM5 are also consistent in depicting the seasonal variation of the trend.

5. On the contrary, both MODIS retrievals and CAM5 simulations display the increasing $\tau_c$ trends over the tropical Indian Ocean/Bay of Bengal (IND), and the Arabian Sea (ARB), due to the influence of increased anthropogenic emissions from South Asia and to a lesser degree from the Middle East in recent years. The MODIS-based trend of +0.036 decade$^{-1}$ ($p < 0.01$) in the IND outflow region is nearly two times of the CAM5-derived trend ($p < 0.05$). Although the CAM5 $\tau_c$ is higher than the MODIS retrieval by about 48% over the ARB outflow, the two datasets give a similar increasing trend of +0.017 decade$^{-1}$ ($p < 0.01$).

6. The MODIS retrievals show a decreasing trend of -0.021 decade$^{-1}$ ($p < 0.05$) and -0.023 decade$^{-1}$ ($p < 0.01$) for combustion AOD in the northwestern Pacific Ocean (NWP) and tropical western Pacific Ocean (TWP), respectively, both being influenced by the anthropogenic emissions from East Asia. The decreasing trend appears to be stronger after 2008 (NWP) and 2005 (TWP), consistent with the transition from an increase to a decrease of SO$_2$ emissions from China as documented in previous studies. The MODIS retrievals also show a decreasing trend of -0.012 decade$^{-1}$ ($p < 0.05$) for dust AOD in the NWP outflow region, which is consistent with the detected decreasing trends of dust emission and storm frequency over China. However, the CAM5 simulations do not reproduce these declining trends.

7. In other outflow regions strongly influenced by biomass burning smoke or dust, neither MODIS retrievals nor CAM5 simulations show statistically significant trends, arguably due to the episodic nature of smoke and dust emissions. The MODIS observed interannual variability is generally larger than the CAM5 simulations. It is also found that the CAM5-MODIS difference in aerosol optical depth is in general significantly larger in smoke and dust outflow regions than those affected mainly by fossil-fuel emissions, highlighting the difficulty in quantifying the episodic dust and smoke emissions and their evolution.

The aerosol trend detection is restricted by several issues associated with satellite remote sensing, such as instrument calibration, limited sampling obscured by the presence of clouds and sun glints, and various assumptions in the retrieval algorithms (Li et al., 2009). To minimize the influences of these limitations on the trend analysis, we have used the MODIS/Aqua data products that are derived from consistent retrieval algorithms with robust calibrations. Long-term data records are required to aggregate sufficient sampling (Chedin et al., 2018), with the required length of data record depending on region. In downwind outflow regions of North American and West European combustion sources, the downward trends are statistically significant because of continuous emission reductions. The upward trends over the tropical Indian Ocean and



Arabian Sea are also statistically significant due to the continuous increase of combustion emissions. For these regions the length of data record might not be very critical and the 15-years of data used in this study deems to be adequate. However, in regions where emissions have undergone non-monotonous changes or monotonous emissions trends are entangled with meteorological variability in a complex way, the 15-years of data is not adequate. We showed that MODIS observations over

the western Pacific Ocean display a less statistically significant decreasing trend of combustion aerosol than that over tropical Indian Ocean and Arabian Sea, because the emissions in East Asia particularly in China have increased and then decreased over the last two decades as a result of economic development and implementation of air pollution control regulations. We also did not detect statistically significant trends for the majority of dust and smoke outflow regions, arguably because sporadic emissions of dust and smoke and their transport are strongly modulated by meteorological conditions. Although Chedin et al.

(2018) suggest that 14-15 years of IASI data record would be adequate to examine the dust trend over Africa, the use of a 15-year MODIS data set in this study did not yield a trend of dust near the coast of North Africa. This discrepancy may have resulted from inherent differences in satellite sampling and instrument sensitivity, among others. Clearly, a longer MODIS data record is required to reliably detect any trends of smoke and dust aerosol.

Our analysis shows that the interannual variability of combustion AOD in major outflow regions is largely consistent with that of emissions from upwind continental sources. Although this finding manifests the importance of correctly characterizing the variability of combustion emissions, roles of meteorology in transporting aerosol from combustion sources downwind to outflow regions cannot be ruled out. For dust and smoke aerosol, both their emissions and transport are strongly modulated by meteorology and hence their variability and trend must account for meteorological factors. Well-designed model sensitivity

experiments are needed to disentangle meteorological variability from that of emissions.

In this study, we have categorized satellite observed total aerosol into three broad types, namely marine aerosol, combustion aerosol, and dust. We have shown that in comparison to an analysis of total AOD, this broad categorization has provided additional insights into the variability and trend of aerosol in terms of their sources, which is useful for guiding model

improvement. Given the complexity of aerosol sources and transformation processes, such the broad aerosol categorization is still not adequate for fully understanding the aerosol trend and variability. For example, the "combustion" aerosol has been loosely defined in this study, because it does not separate aerosol derived from fossil-fuel and biomass burning, and sulfate derived from industrial and volcanic activities. Although some effort is emerging in attempting retrievals of aerosol chemical composition from satellites [Li et al., 2019], accurate and detailed characterization of chemical composition will most likely

rely on an accumulation of in situ observations from sub-orbital platforms.

**Data availability:** The MODIS Dark Target aerosol data are obtained from the NASA Level-1 and Atmosphere Archive and Distribution System (LAADS) webpage (https://ladsweb.nascom.nasa.gov/). The derived MODIS combustion aerosol and dust



AOD data are archived in NASA GSFC clusters and personal computers. The CAM5 simulations were performed and archived
at DOE's National Energy Research Scientific Computing Center (NERSC).

**Author contributions:** HY perceived the project and derived the MODIS combustion and dust AOD with help and input from QT, MC, LAR, and RCL. YY, HW, and SJS designed and ran the CAM5 simulations. HY and YY led the satellite and model integrated analysis and wrote first draft of the paper. All other co-authors participated in discussions of data analysis and
revised the paper.

**Competing interests:** The authors declare that they have no conflict of interest.

**Acknowledgements**
This work was supported by the National Aeronautics and Space Administration's (NASA) Radiation Science Program, CALIPSO/CloudSat Science Team, and Atmospheric Composition Modeling and Analysis Program, managed by Dr. Hal Maring, Dr. David Considine, and Dr. Richard Eckman, respectively, as well as the ACTIVATE project (a NASA Earth Venture Suborbital-3 investigation) managed through the Earth System Science Pathfinder Program Office. Y.Y., H.W., and S.S. of the Pacific Northwest National Laboratory (PNNL) also acknowledge support by the U.S. Department of Energy (DOE)
Office of Science, Biological and Environmental Research as part of the Earth and Environmental System Modeling (EESM) program. The PNNL is operated for DOE by Battelle Memorial Institute under contract DE-AC05-76RL01830.





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




**Tables**



| Table 1: Characteristic fine-mode fraction for individual aerosol types ($f_c$, $f_d$, and $f_m$) as derived from the MODIS C6 data and GOCART simulations. | | | |
|---|---|---|---|
| Aerosol Type | MODIS/Terra | MODIS/Aqua | GOCART |
| Combustion ($f_c$) | 0.92 | 0.89 | 0.95 |
| Dust ($f_d$) | 0.26 | 0.31 | 0.25 |
| Marine ($f_m$) | 0.55 | 0.48 | 0.78 |





| Table 2: Comparison of characteristic FMF for individual aerosol types as derived from different MODIS data collections (C6, C5, and C4). Values corresponding to C5 and C4 were taken from *Yu et al. (2009).* | | | |
|---|---|---|---|
| Aerosol Type | C6 | C5 | C4 |
| Combustion ($f_c$) | 0.92 | 0.90 | 0.92 |
| Dust ($f_d$) | 0.26 | 0.37 | 0.51 |
| Marine ($f_m$) | 0.55 | 0.45 | 0.32 |








**Table 3: Trends of combustion AOD (decade$^{-1}$) with seasonal distinction derived from the MODIS/Aqua retrievals and CAM5 simulations in NAT and MED outflow regions. A trend with significance level of $p < 0.05$ and $p < 0.01$ is marked with $^*$ and $^{**}$, respectively. A trend without asterisk mark is considered not statistically significant.**

| Outflow region | NAT | | MED | |
|---|---|---|---|---|
| | MODIS | CAM5 | MODIS | CAM5 |
| DJF | - 0.005 $^{**}$ | - 0.011 $^{**}$ | - 0.007 | - 0.012 |
| MAM | - 0.021 $^{**}$ | - 0.023 $^{**}$ | - 0.027 $^{**}$ | - 0.022 $^{**}$ |
| JJA | - 0.040 $^{**}$ | - 0.033 $^{**}$ | - 0.025 $^{**}$ | - 0.018 $^{**}$ |
| SON | - 0.010 $^*$ | - 0.012 $^{**}$ | - 0.020 $^*$ | - 0.015 $^{**}$ |
| Annual | - 0.019 $^{**}$ | - 0.020 $^{**}$ | - 0.020 $^{**}$ | - 0.017 $^{**}$ |

**Table 4: Similar to Table 3 but for trend of combustion AOD (decade$^{-1}$) in IND and ARB outflow regions. A trend with significance level of $p < 0.05$ and $p < 0.01$ is marked with $^*$ and $^{**}$ respectively. A trend without asterisk mark is considered not statistically significant.**

| Outflow region | IND | | ARB | |
|---|---|---|---|---|
| | MODIS | CAM5 | MODIS | CAM5 |
| DJF | + 0.050 $^{**}$ | + 0.047 $^{**}$ | + 0.025 $^{**}$ | + 0.031 $^*$ |
| MAM | + 0.062 $^{**}$ | + 0.028 $^*$ | + 0.023 $^{**}$ | + 0.026 $^{**}$ |
| JJA | + 0.003 | -0.002 | - 0.009 | - 0.001 |
| SON | + 0.034 $^*$ | +0.013 | + 0.031 $^{**}$ | + 0.014 |
| Annual | + 0.036 $^{**}$ | +0.020 $^*$ | + 0.017 $^{**}$ | + 0.017 $^{**}$ |




**(a) Outflow regions**

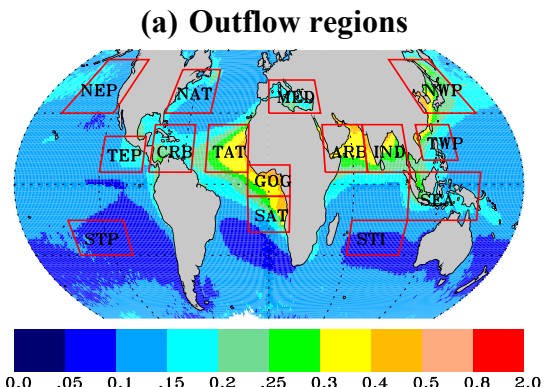

**(b) Source regions**

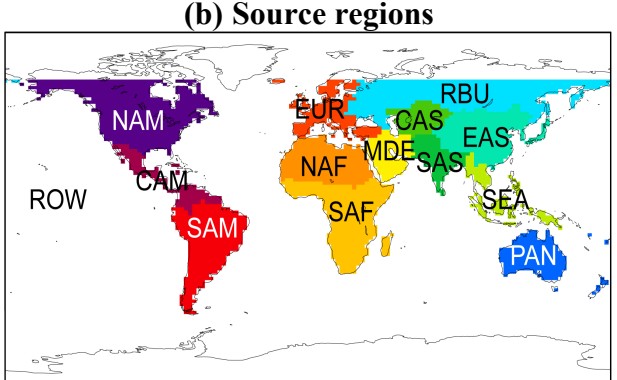

**(c) Source attribution of combustion AOD in the outflow regions**

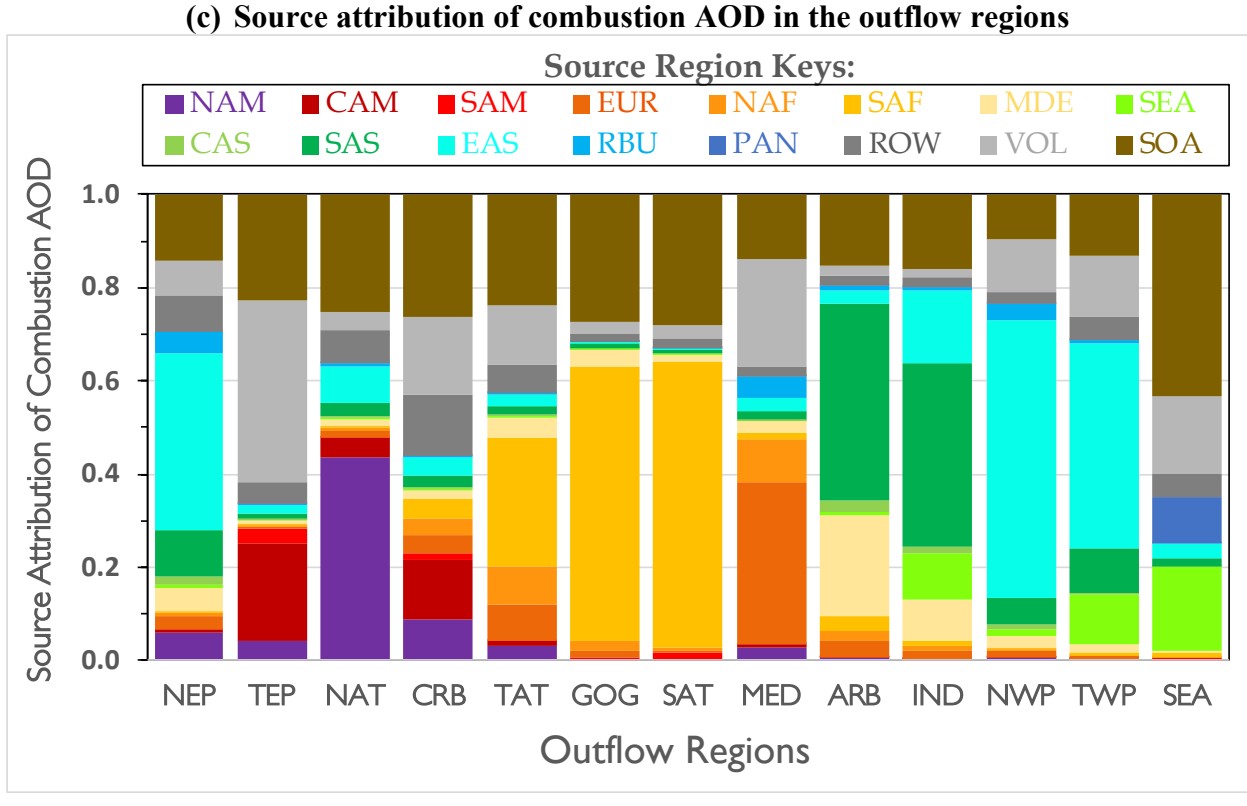

**Figure 1: Illustration of (a) 13 continental outflow regions (covering only water-body portion of boxes) plus two remote regions (STI and STP) overlaid a map of MODIS AOD climatology, (b) 14 source regions used in the CAM5 tagged simulations, and (c) CAM5 source attributions of combustion AOD in the outflow regions based on 2003-2015 simulations.**



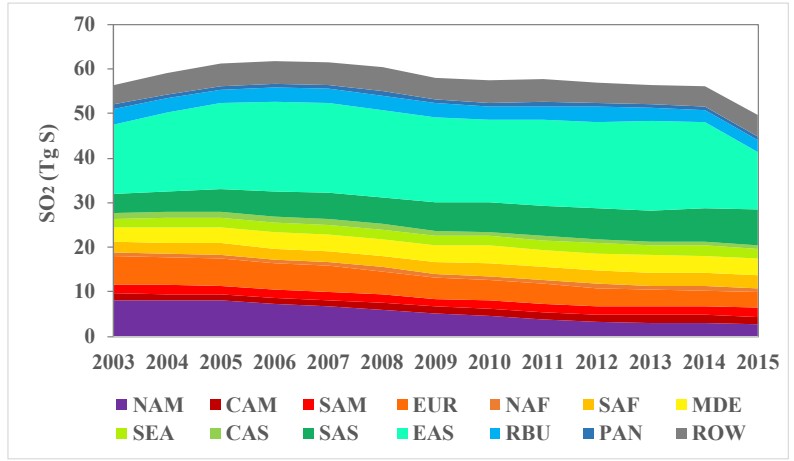

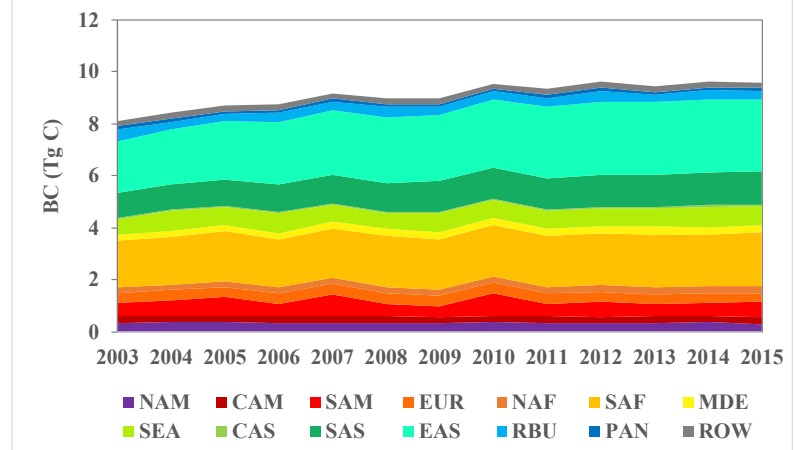

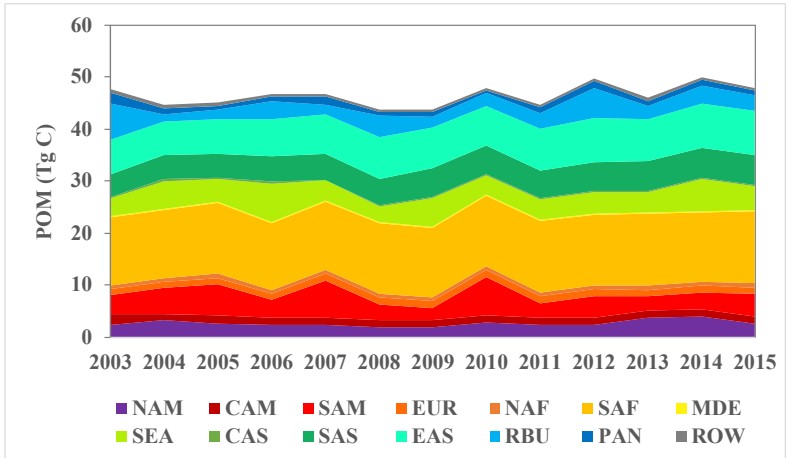





**Figure 2: 2003-2015 source-region stacked annual emissions of SO₂, BC, POM used in the CAM5 model.**

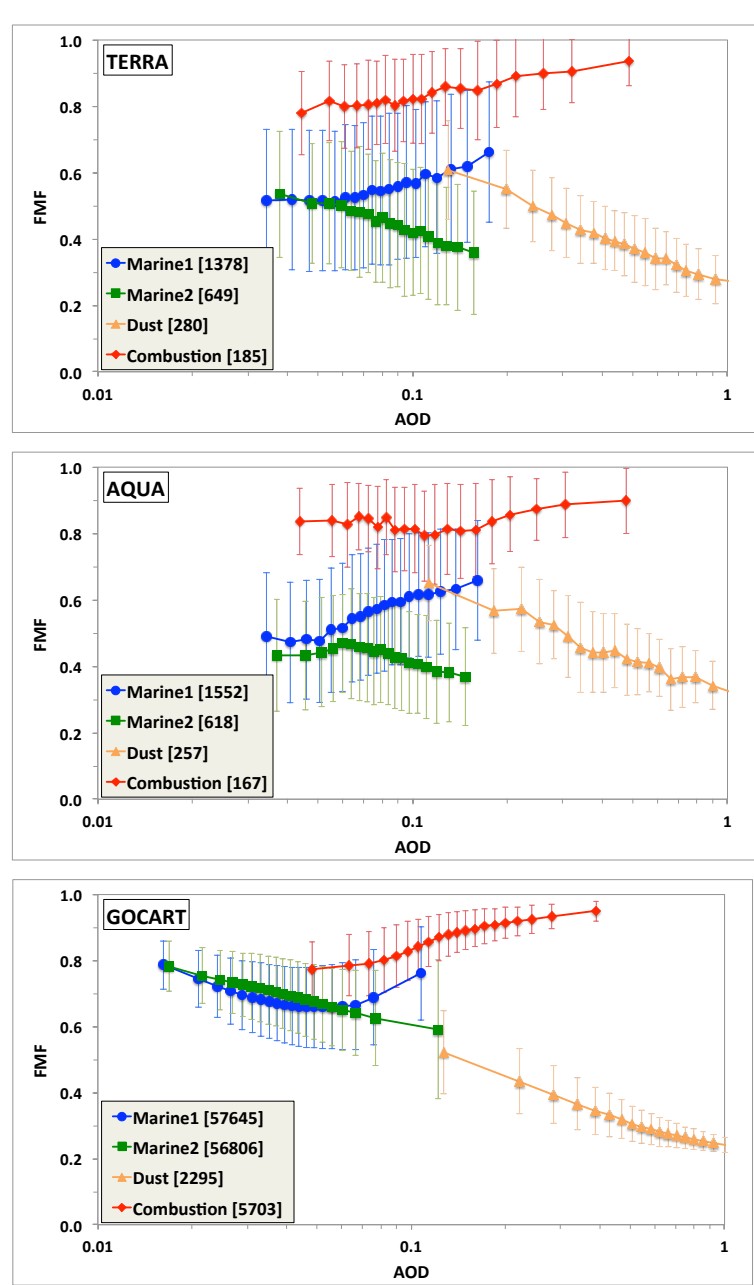

**Figure 3: Aerosol fine-mode fraction (FMF) as a function of AOD in three regions representative of marine aerosol, dust, and combustion aerosol, derived from the MODIS observations on Terra (top panel) and Aqua (middle panel) and the GOCART model**
**(bottom panel). All the data are sorted into 20 AOD bins with equal data points (numbers in brackets) and mean (marked with symbol) and standard deviation (vertical bar) of FMF are calculated.**



**Figure 4: 2003-2015 AOD climatology (top panels) and its partition into combustion, dust, and marine aerosol as derived from MODIS/Aqua observations (left panels) and CAM5 simulations (right panels). Note that color scale for total AOD (top panels) is different from that for its components.**





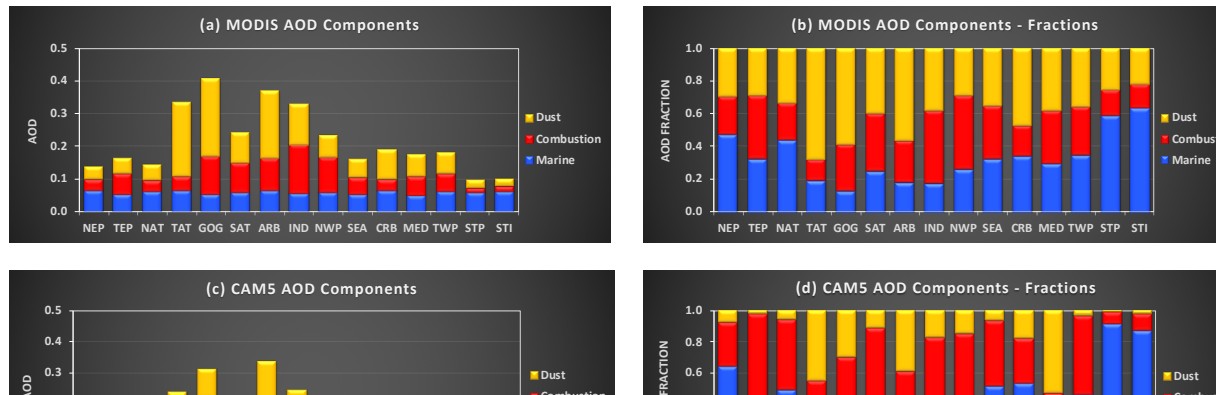

**Figure 5: A climatology of aerosol components (i.e., marine, combustion, and dust) in 15 selected regions (see Figure 1a for definition) based on the MODIS/Aqua 2003-2017 observations: (a) AOD, (b) AOD fraction, and the CAM5 2003-2015 simulations: (c) AOD, (d) AOD fraction.**

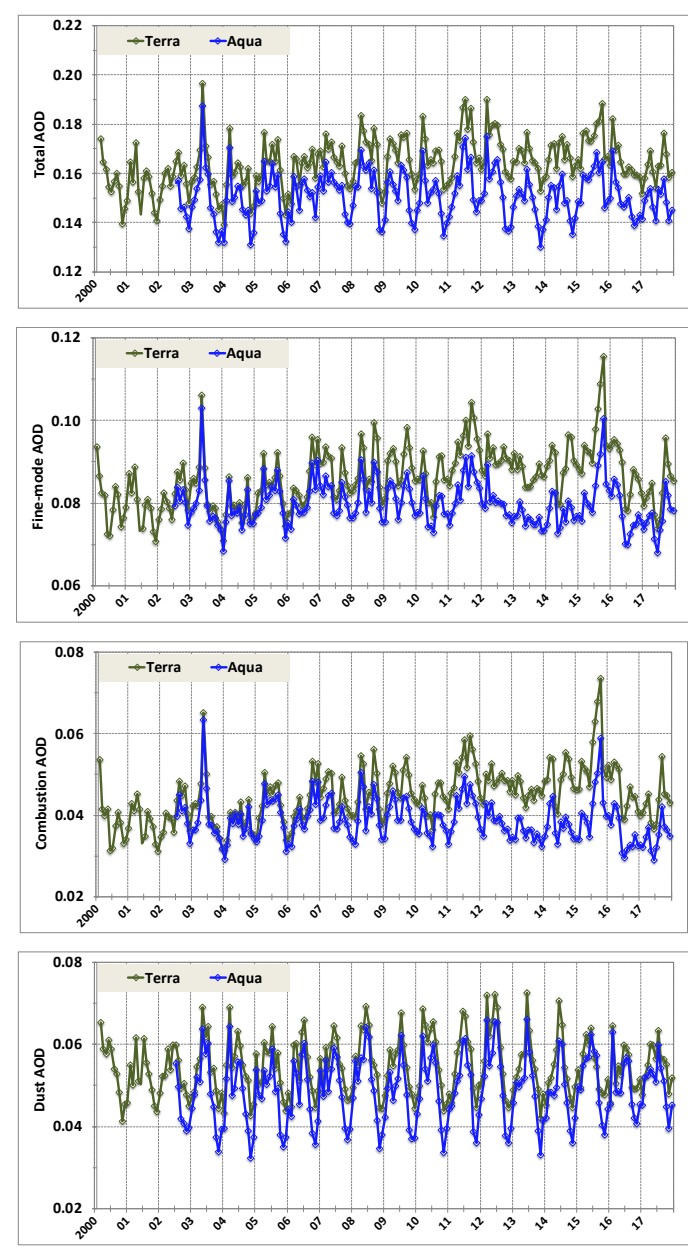


Figure 6: comparison of global-ocean, monthly average total AOD (1st panel), fine-mode AOD (2nd panel), combustion AOD (3rd panel), and dust AOD (4th panel) derived from the MODIS/Terra (gree) and MODIS/Aqua (blue) observations over the 2000-2017 period.






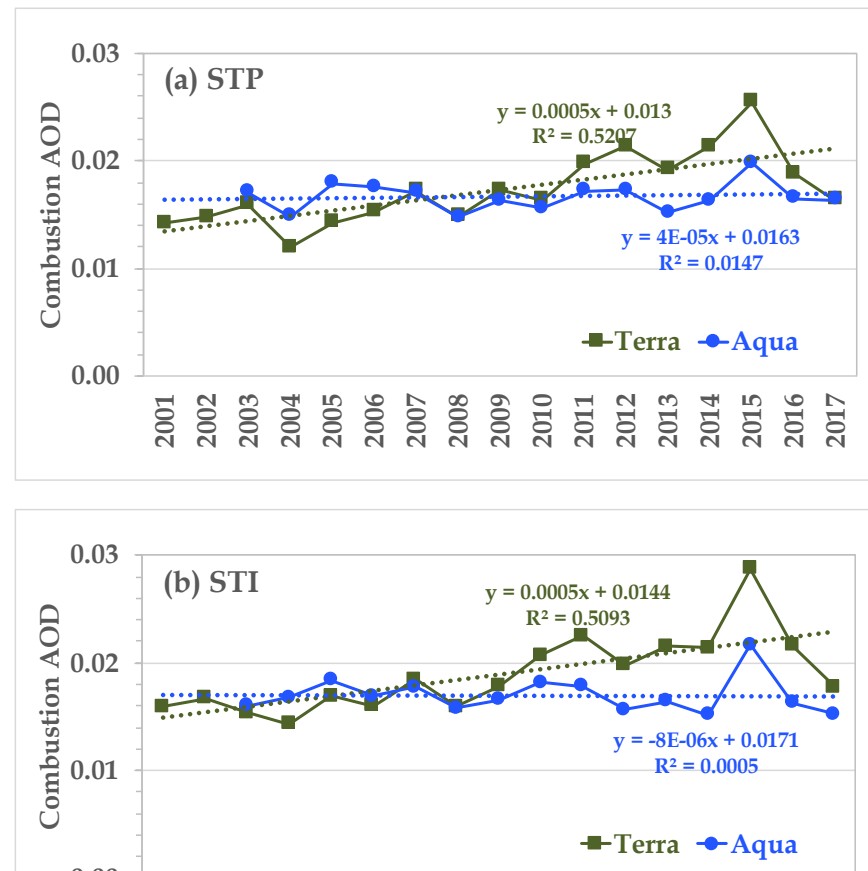

**Figure 7: MODIS retrievals of $\tau_c$ in (a) the southern tropical Pacific Ocean (STP) and (b) the southern tropical Indian Ocean (STI). While Aqua retrievals show no trend, Terra retrievals indicate a statistically significant ($p < 0.001$) and increasing $\tau_c$ trend of 0.005 decade$^{-1}$.**






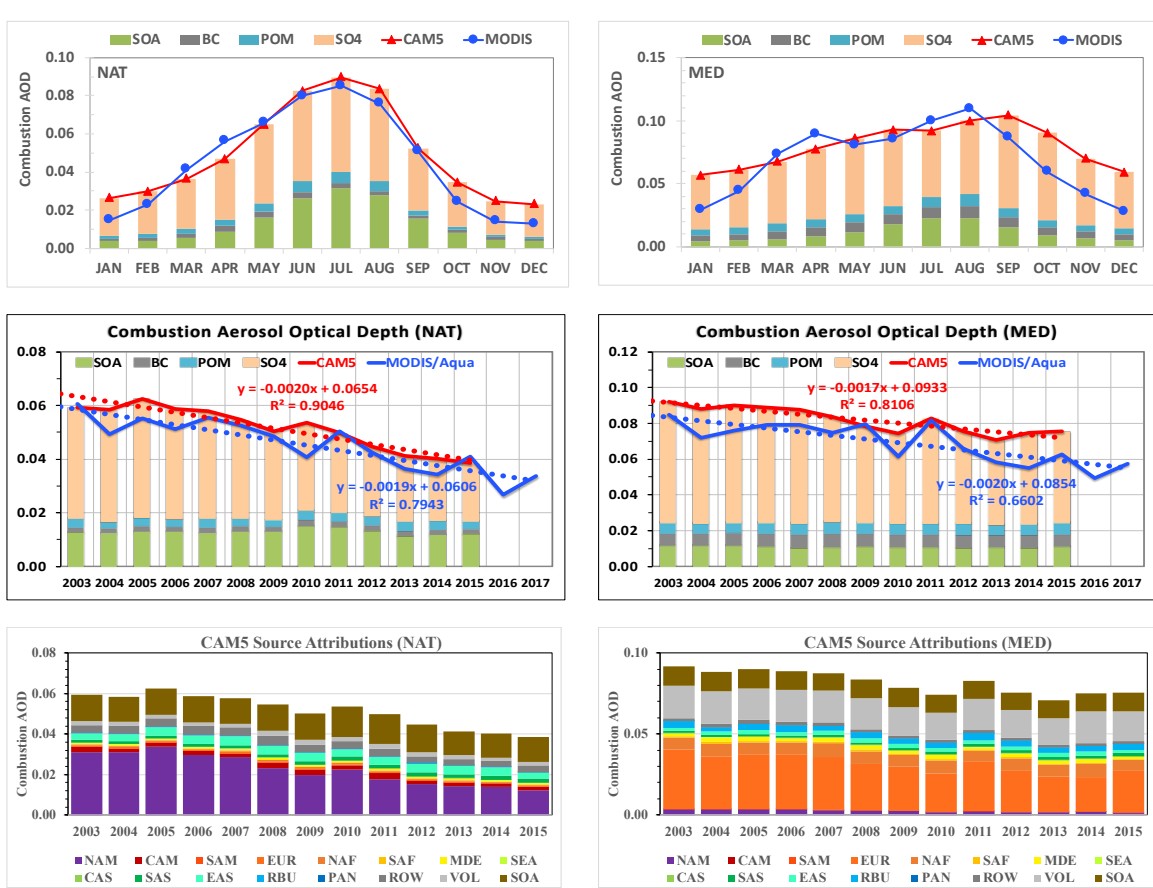

**Figure 8: Climatological seasonal cycle of $\tau_c$ (top panels), interannual variability (solid line) and linear trend (dotted line) (middle panels) retrieved by the MODIS/Aqua (blue) and simulated by the CAM5 model (red) in the North Atlantic Ocean (NAT, left panels) and the Mediterranean Sea (MED, right panels) outflow regions. In both panels, stacked bar shows components (SO₄, POM, BC, SOA) of CAM5 $\tau_c$. The bottom panels show the source attributions of $\tau_c$ simulated the CAM5 model.**




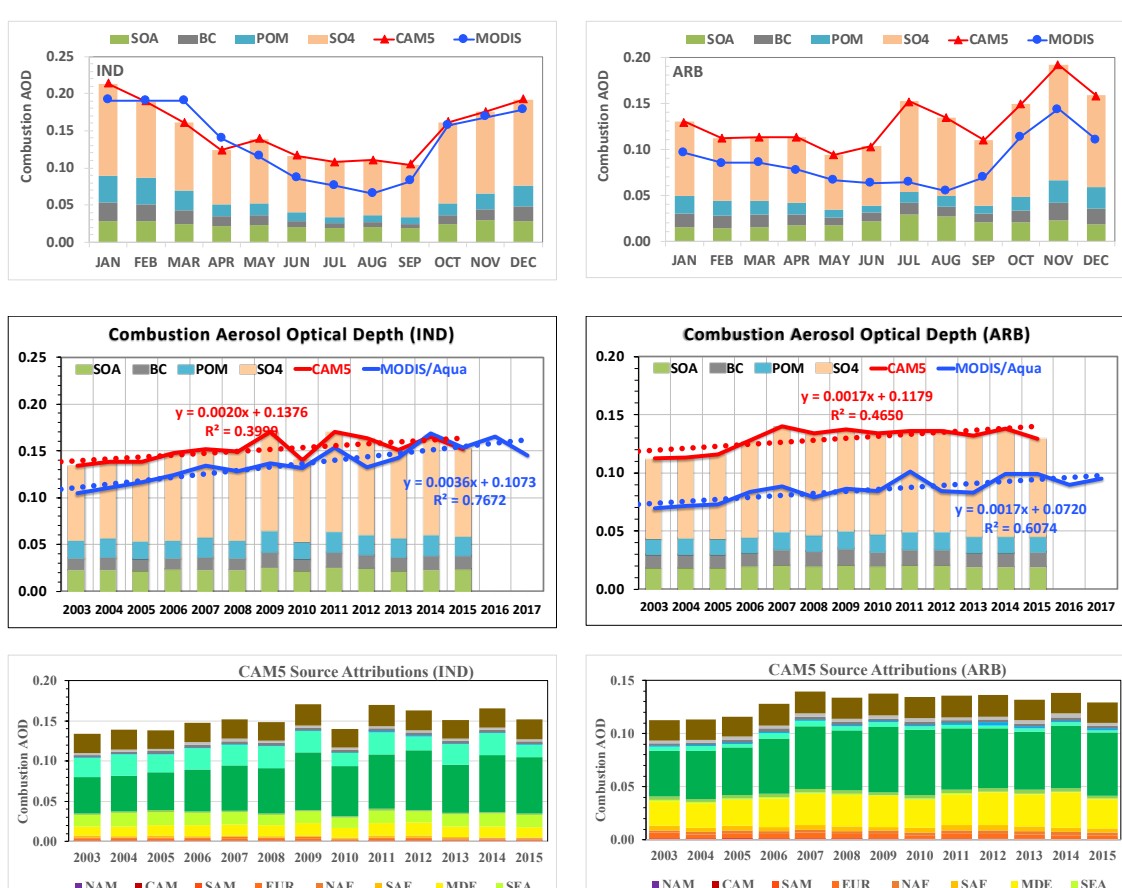

**Figure 9: Same as Figure 8 but for the tropical Indian Ocean/Bay of Bengal (IND, left panels) and the Arabian Sea (ARB, right panels) outflow regions.**




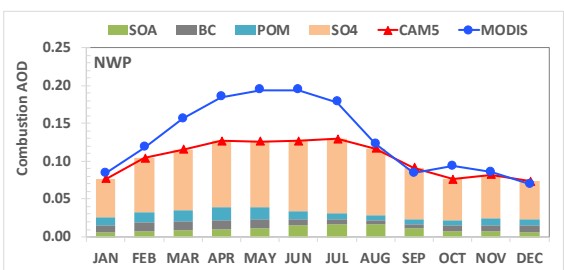

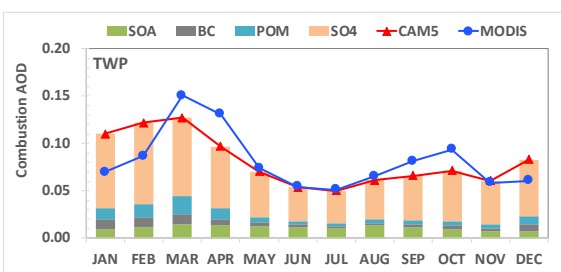

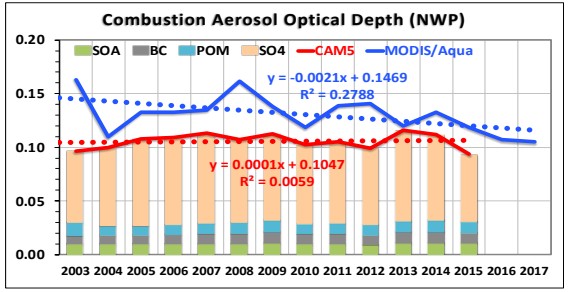

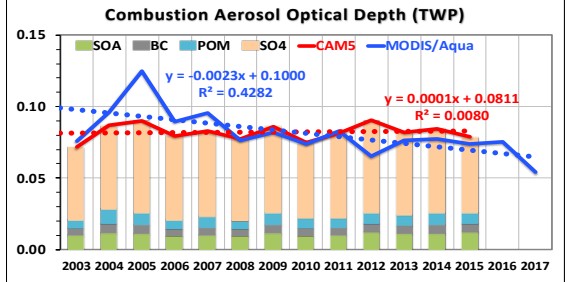

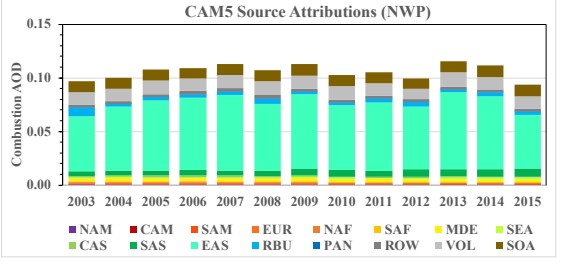

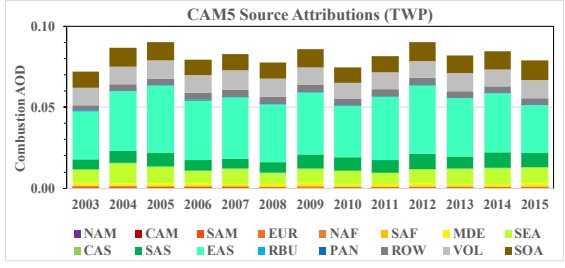

**Figure 10: Same as Figure 8 but for the northwestern Pacific Ocean (NWP, left panels) and the tropical western Pacific (TWP, right panels) outflow regions.**






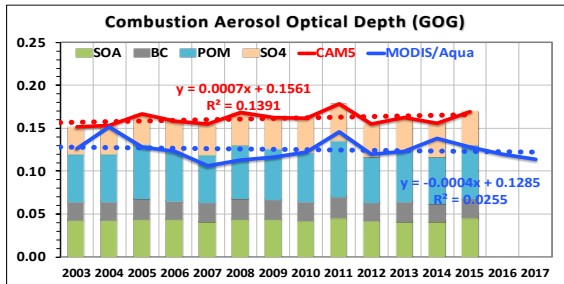

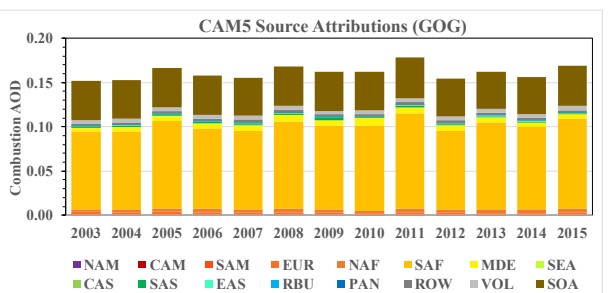

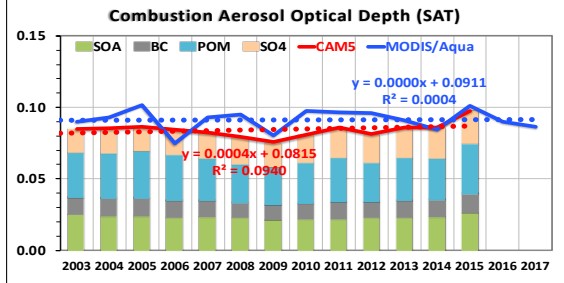

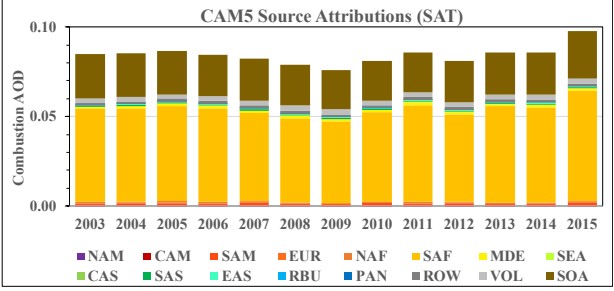

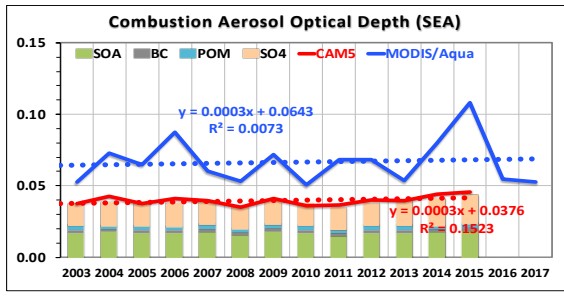

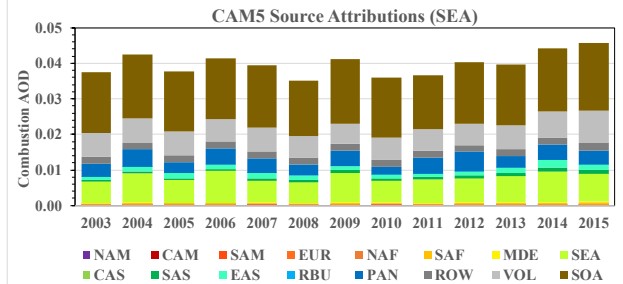

**Figure 11: Interannual variability of $\tau_c$ (left panels) in three outflow regions strongly influenced by biomass burning smoke (the Gulf of Guinea – GOG, the southeastern Atlantic Ocean - SAT, and the southeastern Asia - SEA), as revealed by the MODIS/Aqua retrievals (blue lines) and the CAM5 simulations (red lines). Stacked bar shows the components (SO$_4$, POM, BC, and SOA) of**
**CAM5 $\tau_c$. The CAM5 source attributions are shown in corresponding right panels.**





**Figure 12: Interannual variability of $\tau_d$ in six major outflow regions (the tropical Atlantic Ocean - TAT, the Mediterranean Sea - MED, the Caribbean Basin - CRB, the northwestern Pacific Ocean - NWP, the Arabian Sea - ARB, and the tropical Indian Ocean and the Bay of Bengal – IND) as revealed by the MODIS/Aqua retrievals (blue lines) and CAM5 simulations (red lines).**