# Peer review of "Interannual Variability and Trends of Combustion Aerosol and Dust in Major Continental Outflows Revealed by MODIS Retrievals and CAM5 Simulations During 2003 – 2017"

_Atmospheric Chemistry and Physics, 2019_

## Referee Comment (RC1) · Anonymous Referee #3 · 27 Sep 2019

The authors present an evaluation of model and satellite observations of mineral dust/combustion aerosols over a fifteen -year period. They discuss the strengths/weaknesses of the MODIS products and various versions of the CAM simulations. In this manuscript further highlight the need to consistently use MODIS data to avoid errors, and the MODIS/Terra data are not to be used for AOD trends. Further, the conclusions regarding dust model simulations needing significant improvement are encouraging.

Overall, the manuscript is a challenging read due to the flamboyant reference of statis-

tics. It is understood that the authors are putting the current evaluation in context of previous work, but it significantly impacts the readability of the manuscript. Other than the manuscript's readability, my comments on the paper are minor and I believe it should be published after minor revisions.

– Page 6, line 202: Can the authors clarify whether these differences are statistically significant or not?

– After reading the manuscript I could not conclude which (MODIS or CAM5) is more reliable. Page 12, lines 393-398 and section 3.3.3 left me somewhat confused. Can the authors clarify which is a "better" product according to their research?

– Page 17, line 561: I am confused regarding the use of "monotonous" here. Please consider changing.

– Page 18: line 583: "...processes, such the broad..." should be "...processes, such a broad..."?

– Figure 6 caption: Capitalize first word in sentence.

---

## Referee Comment (RC2) · Anonymous Referee #2 · 30 Sep 2019

General comments:

In the paper, "Interannual Variability and Trends of Combustion Aerosol and Dust in Major Continental Outflows Revealed by MODIS Retrievals and CAM5 Simulations During 2003-2017", the authors analyze both satellite-based and model-based datasets of various aerosol types in continental outflow regions over a fifteen-year period. A great deal of research has been completed and the narrative is generally well-written. I find no significant issues with this study; however, I believe it can be improved in a few areas. Several technical errors were found, and while I outline many of them in my review, the

paper should be given a thorough round of proofreading edits. Overall, I recommend a minor revision for this manuscript. The authors should address the specific comments and technical corrections discussed below.

Specific comments:

1. Page 4, Line 12: Are you using the most recent MODIS data available (i.e., Collection 6.1)?

2. Page 5, Line 155: Why is SOA not included in the tagging?

3. Page 5, Lines 162-163: It does not look like an increase to me. Please double check.

4. Page 7, Lines 220-222: What is the uncertainty associated with this?

5. Page 8, Line 231: Do you have sources/citations for the claim that the derived dust AOD is likely overestimated?

6. Page 8, Line 244: In addition to AERONET, you can also compare against Maritime Aerosol Network (MAN) observations (since here much of the discussion is focused on over ocean)?

7. Page 8, Line 249: Please make sure this is mentioned somewhere later in the paper.

8. Page 8, Line 255: Be clearer here with your definition of fractional AOD. Also, in Figures 5 (b) and (d), I would not show AOD fraction this way, as it can be confusing. For example, dust always appears to be at 100%. I suggest editing these plots.

9. Page 10, Lines 301-304: Include the values of each trend.

10. Figure 1: I thought SOA was not included in the tagging? Also, is the y-axis supposed to be in percentages? Please add the correct units.

11. Figure 2: Shouldn't the units here for the y axes be just Tg?

12. Figure 6: Add the trend lines and trends for each plot.

13. Figure 8: Why no CAM5 for 2016-2017? Is the MODIS trend only up to 2015, or does it include 2016-2017? It is of concern if the trends are computed for different time periods. This same comment can be applied to some of the later figures as well (see below).

14. Figures 9, 10, and 11: same concerns/questions/comments as Figure 8.

15. Figure 11: How come no climatological seasonal cycle plots are shown, as in Figs. 8-10?

16. Figure 12: Are you comparing trends from different time periods?

Technical corrections:

1. Page 4, Line 120: "sea-spay" should be "sea-spray".

2. Page 9, Line 268: The "0.025" here should be "0.015", correct?

3. Page 11, Line 340: I would switch "well captures" to "captures well".

4. Page 11, Line 343: "the peak occurs" . . . the peak what? Be specific.

5. Page 11, Line 343: Remove "the" before "MODIS" and "CAM5".

6. Page 11, Line 345: Add "the" before "NAT", "MED", and "NAT and MED regions".

7. Page 11, Line 356: "season" should be "seasons".

8. Page 12, Line 369: Remove "the" before "CALIOP".

9. Page 12, Line 370: Remove "the" before "SeaWiFS".

10. Page 13, Line 405: Remove "the" before "CALIOP".

11. Page 13, Lines 406-407: Define PM2.5, SO2, and CO.

12. Page 13, Line 407: "decline" should be "declining".

13. Page 13, Line 408: "decline" should be "declining".

14. Page 13, Line 409: I would reword this sentence by changing "How" to "The manner in which".

15. Page 13, Line 410: Why is "Figure 10" in bold print?

16. Page 13, Line 417: "possibly" should be "possible".

17. Page 13, Line 422: Remove "the" before "MODIS".

18. Page 13, Line 426: Define ATSR.

19. Page 14, Line 459: Remove "the" before "northern".

20. Page 14, Line 460: Remove "the" before "northern".

21. Page 15, Line 461: Remove "the" before "southern".

22. Page 15, Line 462: Remove "the" before "northern".

23. Page 15, Line 468: In this context, I think "driving up" should be "drove up", although I suggest finding another phrase.

24. Page 15, Line 490: Add "the" before "literature".

25. Page 16, Line 496: Remove "the" before "CALIOP".

26. Page 16, Line 500: Add "of" after "factor".

27. Page 16, Line 506: Collection 6.1, correct?

28. Page 16, Line 507: Replace "the" after "over" with "a".

29. Page 17, Line 528: Remove "only".

30. Page 18, Line 566: Add commas after "Asia" and "China".

31. Page 18, Line 570: Add "the" after "of".

32. Page 18, Line 585: Replace "the" after "such" with "a".

33. Page 26, Table 3 caption: Add "an" before "asterisk".

34. Page 26, Table 4 caption: In the first line, "trend" should be "trends". Add a comma before "respectively". Also, add "an" before "asterisk".

35. Page 27, Figure 1: (a) Add a label to the color bar. (b) Add latitudes and longitudes to the map.

36. Page 28, Figure 2: Add labels (a)-(c) to the plots and the figure caption.

37. Page 29, Figure 3: I suggest using (a)-(c) for labels. Please also edit the caption accordingly. Also, add the difference between the marine cases to the caption.

38. Page 30, Figure 4: Add labels (a)-(h) to the plots and the figure caption.

39. Page 32, Figure 6: Add labels (a)-(d) to the plots and the figure caption.

40. Page 34, Figure 8: Add labels (a)-(f) to the plots and the figure caption. Also, add labels to the y axes of the middle plots. Add "by" after "simulated" in the last line of the figure caption.

41. Page 35, Figure 9: Add labels (a)-(f) to the plots and the figure caption. Also, add labels to the y axes of the middle plots.

42. Page 36, Figure 10: Add labels (a)-(f) to the plots and the figure caption. Also, add labels to the y axes of the middle plots.

43. Page 37, Figure 11: Add labels (a)-(f) to the plots and the figure caption. Also, add labels to y axes of the plots on the left-hand-side of the figure.

44. Page 38, Figure 12: Label the x and y axes for each plot. Also, add labels (a)-(f) to the plots and the figure caption. In the figure caption, remove "and" before "the tropical Indian Ocean".
* * *

---

## Referee Comment (RC3) · Anonymous Referee #1 · 2 Oct 2019

In this study, the authors examined the 15-year trends and interannual variabilities of dust and combustion aerosols using MODIS retrievals from 2003 to 2017, with the aid of CAM5 simulation. The goal of this paper is clearly stated in the text, and scientifically important. The tables and figures are well prepared. The approaches used is well-established. I recommend publication of this paper with several minor modifications.

General comments:

1. The term of "combustion aerosol" may not be suitable for SOA (shown as green

bars in Figures 8~11), since some SOA, like biogenic SOA, is not formed through combustion processes. The authors can either use term "pollution" as in Yu et al., (2009), or explicitly state what aerosol species in CAM5 are considered as "combustion aerosols".

2. Why dust emissions from different regions are not tagged? From Figure 4, it seems that some regions are very likely affected by dust emitted from different regions. It may also help to diagnose the discrepancies between MODIS and CAM5 simulations as shown in Figure 12.

3. In Figure 8~Figure 11, what types of simulated sulfate are considered? I am assuming sulfate formed from DMS and sulfate formed in coarse mode are excluded in the plots. Is this correct?

4. Strictly speaking, the presence of clouds affects MODIS retrievals of aerosols, but not the CAM5 simulations. Is cloud screening performed for CAM 5 analysis also?

Specific comments:

Line 135: please mention the exact version of CAM5 used in the study.

Line 142: I think it is called CEDS emission dataset in Hoesly et al. (2018), and the dataset is only available till 2014? Is this correct?

Line 149: By tagging SO2, the source regions of sulfate aerosols can also be tracked. Is this correct? If so, please mention it in the text.

Line 196 and Table 1: Are fc and fd derived from Figure 3 used for all 13 outflow regions? If I remember correctly, in Yu et al. (2009), different sets of fc and fd for different regions and seasons are derived. This is important since different aerosol characteristics in different regions and seasons.

Line 336 "simulated relative contributions by sulfate, POM, BC, and SOA to the total AOD". Should it be total AOD or ïĄťc only, because the authors are trying to compare

combustion AOD here. We know that, in MAM3/CAM5, SO2 can condense on accumulation mode and coarse mode and form sulfate at the same time. Are sulfate aerosols in coarse mode considered as one contributor of Tau c or Tau d?

Line 416: The spherical dust assumption may explain the large difference in spring since it is dust storm season in China. However, it can not explain the large difference in June and July, since the occurrence frequency of dust storm in these two months are not high.

Figure 10: It is well known that anthropogenic aerosol concentrations peaks in winter season in EAS region (or China), like in Zhang et al. (2012, https://doi.org/10.5194/acp-12-779-2012). And it is well known that CAM5 fails to reproduce observed seasonality of sulfate aerosols in China. Therefore, it surprises me to see that combustion AOD in NWP does not peak in winter. What are the reasons?

Figure 12. and Line 483: As shown in the figure, it appears to me that the interannual variability in CAM5 simulation is much smaller compared to observations. What is the reason? How comes the nudged simulation can not reproduce observed interannual variability?

---

## Author Comment (AC1) · 8 Nov 2019

Referee #1 In this study, the authors examined the 15-year trends and interannual vari­abilities of dust and combustion aerosols using MODIS retrievals from 2003 to 2017, with the aid of CAM5 simulation. The goal of this paper is clearly stated in the text, and scientifically important. The tables and figures are well prepared. The approaches used is well established. I recommend publication of this paper with several minor modifications.

[Figure]

Response: We appreciate the reviewer's effort of helping us improve the manuscript. Below is our point-by-point response to the comments. And we will revise the paper accordingly and submit the revised paper for further comments.

General comments: 1. The term of "combustion aerosol" may not be suitable for SOA (shown as green bars in Figures 8_11), since some SOA, like biogenic SOA, is not formed through combustion processes. The authors can either use term "pollution" as in Yu et al., (2009), or explicitly state what aerosol species in CAM5 are considered as "combustion aerosols".

You are definitely right that the "combustion" aerosol is not always suitable for SOA because of contributions by biogenic SOA in some regions. So is the "pollution" aerosol. We changed the early use of "pollution" aerosol to "combustion" aerosol, considering that mineral dust is often referred to as "pollution". The word "combustion" can at least exclude the dust.

We agree that we should state explicitly what aerosol species in CAM5 simulations are considered as "combustion" aerosol. This is done in line 236-239, page 8. The combustion AOD is "a sum of AOD of SO4 (excluding those generated from DMS chemistry), BC, POM, and SOA".

2. Why dust emissions from different regions are not tagged? From Figure 4, it seems that some regions are very likely affected by dust emitted from different regions. It may also help to diagnose the discrepancies between MODIS and CAM5 simulations as shown in Figure 12.

The natural dust and sea salt emissions are calculated online in the model. They are different from anthropogenic aerosol emissions that are provided by offline emission files. We agree with the referee that it would be interesting to diagnose dust source attribution as well, but the dust tagging, which requires additional coding, is not available in the current CAM5 model. We do plan to add the dust source tagging in the future model development.

3. In Figure 8_Figure 11, what types of simulated sulfate are considered? I am assuming sulfate formed from DMS and sulfate formed in coarse mode are excluded in the plots. Is this correct?

Sulfate in Figures 8-11 includes those from anthropogenic sources, such as industry, power plants, agricultural, residential, international shipping, surface transportation, and waste treatment emissions, open biomass burning emissions and volcanic eruptions. Sulfate formed from DMS is excluded here. Sulfate formed in coarse mode is included if it is not formed from DMS.

4. Strictly speaking, the presence of clouds affects MODIS retrievals of aerosols, but not the CAM5 simulations. Is cloud screening performed for CAM 5 analysis also?

Yes, MODIS retrievals are made in cloud-free conditions. For the CAM5 simulations, we did not perform cloud screening, because the coarse resolution of CAM5 makes it difficult to identify meaningful number of cloud-free grids. In addition, CAM5 clouds could be different from MODIS clouds, which may still complicate the comparison.

Specific comments: Line 135: please mention the exact version of CAM5 used in the study.

CAM5.3 was used in this study. It is now mentioned in the manuscript.

Line 142: I think it is called CEDS emission dataset in Hoesly et al. (2018), and the dataset is only available till 2014? Is this correct?

Correct. CEDS historical anthropogenic emission dataset from 1979ň–2014 (version 20160726) and open biomass burning emissions from 1979–2015 (version 20161213) are used in this study. For anthropogenic emissions in year 2015, emission data are interpolated from SSP (Shared Socioeconomic Pathways) 2-4.5 forcing scenario (Riahi et al., 2017).

Riahi, K., et al.: The Shared Socioeconomic Path ways and their energy, land use, and greenhouse gas emissions implications: An overview, Global Environ. Chang., 42,

153–168, https://doi.org/10.1016/j.gloenvcha.2016.05.009, 2017.

Line 149: By tagging SO2, the source regions of sulfate aerosols can also be tracked. Is this correct? If so, please mention it in the text.

Yes. The sentence has been revised to "Aerosols and their precursor emissions, including SO2, sulfate, BC and POM, are tagged with respect to 14 source regions."

Line 196 and Table 1: Are fc and fd derived from Figure 3 used for all 13 outflow regions? If I remember correctly, in Yu et al. (2009), different sets of fc and fd for different regions and seasons are derived. This is important since different aerosol characteristics in different regions and seasons.

We used Figure 3 derived fc and fd for all regions without accounting for their spatial and seasonal variations. This is similar to Yu et al. (2009). We don't have adequate data to refine this assumption. But for marine aerosol fine-mode fraction fm, we do consider seasonal and spatial variations following the method described in Yu et al. (2009).

Line 336 "simulated relative contributions by sulfate, POM, BC, and SOA to the total AOD". Should it be total AOD or ïAËŽt'c only, because the authors are trying to compare combustion AOD here. We know that, in MAM3/CAM5, SO2 can condense on accumulation mode and coarse mode and form sulfate at the same time. Are sulfate aerosols in coarse mode considered as one contributor of Tau c or Tau d?

The sentence has been revised to "simulated relative contributions by sulfate, POM, BC, and SOA to the total combustion AOD". Sulfate aerosols in coarse mode of CAM5 are also considered as one contributor of combustion AOD. Note that sulfate converted from DMS is excluded here.

Line 416: The spherical dust assumption may explain the large difference in spring since it is dust storm season in China. However, it can not explain the large difference in June and July, since the occurrence frequency of dust storm in these two months

are not high.

We agree and the cloud contamination may be a major reason for the difference in June and July, We have rephrased the sentences: "It is thus possible that the large MODIS-CAM5 difference during the dust season (March-May) could at least be partially attributed to the spherical dust assumption in the MODIS algorithm. The higher MODIS ïĄťc in June and July is likely a result of cloud contamination and limited sampling."

Figure 10: It is well known that anthropogenic aerosol concentrations peaks in winter season in EAS region (or China), like in Zhang et al. (2012, https://doi.org/10.5194/acp-12-779-2012). And it is well known that CAM5 fails to reproduce observed seasonality of sulfate aerosols in China. Therefore, it surprises me to see that combustion AOD in NWP does not peak in winter. What are the reasons?

We agree with the reviewer that anthropogenic aerosol concentrations near the surface in eastern China peak in winter. However, what we are looking at in this study is the column combustion AOD in East Asia outflow region. Thus at least two factors could contribute to the difference in seasonality. First, columnal AOD and surface concentration can differ in seasonality, because of the seasonal variation of the mixing layer or the atmospheric boundary layer (ABL). Aerosol is mixed within much shallower ABL in winter than in other seasons, leading to maximum surface PM concentration in China. Second, the seasonality of aerosol outflow over ocean can be largely affected by the aerosol transport and removal processes. For example, aerosol outflow from East Asia has been found to peak in spring when strong ascending airstreams lift pollutants into the free troposphere and then the westerlies carry them across the Pacific (Yang et al., 2015).

With regard to model simulations of sulfate, the default CAM5 using IPCC AR5 emissions failed to reproduce observed seasonality of sulfate aerosols in China, likely in part because IPCC AR5 emissions do not have seasonal variability in anthropogenic

emissions. In CEDS anthropogenic emissions, seasonal feature of emissions is included, which could have improved the simulated seasonality of sulfate. However, it is possible that the model still has uncertainty in simulating two prerequisites, namely relative humidity and ozone, for sulfate formation as identified in Fang et al. (2019). Detailed analysis of CAM5 sulfate simulation is beyond the scope of this study.

References: Yang, Y., H. Liao, and S. Lou, Decadal trend and interannual variation of outflow of aerosols from East Asia: Roles of variations in meteorological parameters and emissions, Atmos. Environ., 100, 141–153, doi:10.1016/j.atmosenv.2014.11.004, 2015.

Fang, Y., C. Ye, J. Wang, Y. Wu, M. Hu, W. Liu, F. Xu, and T. Zhu, Relative humidity and O3 concentrations as two prerequisites for sulfate formations, Atmos. Chem. Phys., 19, 12295-12307, 2019.

Figure 12. and Line 483: As shown in the figure, it appears to me that the interannual variability in CAM5 simulation is much smaller compared to observations. What is the reason? How comes the nudged simulation can not reproduce observed interannual variability?

Yes, the dust interannual variability in CAM5 simulation is generally much smaller compared to the MODIS observation, except in the Mediterranean region (MED, Figure 12c).

There are many factors that can affect the interannual variability of modeled and observed dust AOD, including biases in model parameterizations of dust emissions, transport, and deposition processes, in reanalysis data used for nudging, and in satellite retrievals. Meteorological nudging used in the CAM5 simulation can only ensure realistic capture of large-scale circulations. Biases in the other aspects of the model could lead to the difference in the mean and interannual variability between model and satellite data.

---

## Author Comment (AC2) · 8 Nov 2019

Referee #2 General comments: In the paper, "Interannual Variability and Trends of Combustion Aerosol and Dust in Major Continental Outflows Revealed by MODIS Retrievals and CAM5 Simulations During 2003-2017", the authors analyze both satellite-based and model-based datasets of various aerosol types in continental outflow regions over a fifteen-year period. A great deal of research has been completed and the narrative is generally well-written. I find no significant issues with this study; however, I believe it can be improved in a few areas. Several technical errors were found, and

while I outline many of them in my review, the paper should be given a thorough round of proofreading edits. Overall, I recommend a minor revision for this manuscript. The authors should address the specific comments and technical corrections discussed below.

We appreciate the reviewer's detailed comments and suggested technical corrections. Below is our point to point response to the comments. The paper will be revised accordingly. Co-author LAR has offered to proofread the paper thoroughly once all comments are addressed.

Specific comments: 1. Page 4, Line 12: Are you using the most recent MODIS data available (i.e., Collection 6.1)?

In this study we used Collection 6, not Collection 6.1. But our major findings will not change if Collection 6.1 is used. Major updates from Collection 6 to 6.1 include (a) the improved calibration (affecting MODIS/Terra more than MODIS/Aqua), (b) updated aerosol models over land, and (c) land/ocean masks. We tested C6 and C6.1 difference by comparing over ocean AOD and FMF from first 15 days of each month in a year. Statistical analysis of C6 versus C6.1 for MODIS/Aqua yields a mean bias of -0.001, RMSE of 0.006, and linear regression of C6 = 0.995 x C6.1 -0.001for AOD. Correspondingly, MODIS/Aqua FMF has a mean bias of 0.005, RMSE of 0.043, and linear regression of C6 = 1.002 x C6.1 + 0.001. Because of the improvement (a), the spurious trend in the MODIS/Terra could have been reduced to some extent in C6.1, but not been removed totally. In this paper we have been using MODIS/Aqua to study the trend and interannual variability. As we emphasize in this and previous studies, it is more important to use data in a self-consistent manner (e.g., characteristic fine-mode fractions for individual components need to be derived from the same data collection). Once the MODIS data are used self consistently, the derived aerosol components agree among different data collections (Yu et al., 2009).

2. Page 5, Line 155: Why is SOA not included in the tagging?

We excluded SOA tagging in the simulation due to the limitation of computational resources and relatively large biases in source attribution from simplified treatments of SOA formation and gas precursor emissions in climate models, including CAM5. The latter makes SOA tagging less useful.

3. Page 5, Lines 162-163: It does not look like an increase to me. Please double check.

The SO2 emissions from South Asia did increase steadily from 4.35 Tg S yr-1 in 2003 to 7.82 Tg S yr-1 in 2015. Because it only contributes a small fraction to the global total, this increase is not clearly seen from the stacked emissions.

4. Page 7, Lines 220-222: What is the uncertainty associated with this?

Although this parameterization of marine aerosol optical depth could introduce significant errors on the basis of regional and short time scales, the derived global mean marine aerosol agrees well with the AERONET-based measurement.

5. Page 8, Line 231: Do you have sources/citations for the claim that the derived dust AOD is likely overestimated?

Although dust plumes originated from the Namibia sources have been observed (Vickery et al., 2013; Formenti et al., 2019), there are no DOD records that can be used to evaluate the MODIS derived DOD in this region. Given that the region has high cloud cover, MODIS retrievals of AOD in particular coarse-mode AOD are prone to cloud contamination. In Yu et al. (2019), we found that MODIS-derived DOD is higher than MISR and IASI DOD south to the equator.

References: Formenti, P., et al., The aerosols, radiation and clouds in southern Africa field campaign in Namibia, Bull. Am. Meteorol. Soc., 100, 1277-1298, 2019.

Vickery, K. J., F. D. Eckardt, and R. G. Bryant, A sub-basin scale dust plume source frequency inventory for southern Africa, 2005–2008, Geophys. Res. Lett., 40, 5274–5279, doi:10.1002/grl.50968, 2013.

[Figure]

Yu, H. et al., Estimates of African dust deposition along the trans-Atlantic transit using the decade-long record of aerosol measurements from CALIOP, MODIS, MISR, and IASI, J. Geophys. Res. -Atmos., 124, 7975-7996, 2019. https://doi.org/10.1029/2019JD030574.

6. Page 8, Line 244: In addition to AERONET, you can also compare against Maritime Aerosol Network (MAN) observations (since here much of the discussion is focused on over ocean)?

Yes, MAN observations with extensive coverage should be useful. For the purpose of evaluate marine aerosol (NOT maritime aerosol), it is necessary to rigorously screen out continental influences from the MAN observations, which is beyond the scope of this study. The MAN observations have been compared with AERONET in-island observations (Smirnov et al., 2009) and used to evaluate MODIS retrievals (e.g., Levy et al., 2013; Remer et al., 2013).

Levy, R. C., et al., The Collection 6 MODIS aerosol products over land and ocean, Atmos. Meas. Tech., 6, 2989–3034, 2013.

Remer, L.A., et al., MODIS 3km aerosol product: algorithm and global perspective, Atmos. Meas. Tech., 6, 1829-1844, 2013.

Smirnov, A., et al., Marine Aerosol Network as a component of Aerosol Robotic Network, J. Geophys. Res., 14, D06204, doi:10.1029/2008JD011257, 2009.

7. Page 8, Line 249: Please make sure this is mentioned somewhere later in the paper.

Yes, we discuss it later in the paper.

8. Page 8, Line 255: Be clearer here with your definition of fractional AOD. Also, in Figures 5 (b) and (d), I would not show AOD fraction this way, as it can be confusing. For example, dust always appears to be at 100%. I suggest editing these plots.

The fraction AOD is defined as fractional contributions of combustion, marine, and dust

aerosol to the total AOD. The sum of three fractions is 1. Figures 5 (b) and (d) show the stacked contributions of the three components distinguished by different colors. We clarify it both in main text and in the figure caption.

9. Page 10, Lines 301-304: Include the values of each trend.

We will consider in revision.

10. Figure 1: I thought SOA was not included in the tagging? Also, is the y-axis supposed to be in percentages? Please add the correct units.

SOA and VOL are not tagged in terms of source regions. But they are accounted for as the combustion aerosol in this study to be consistent with the aerosol component definition for the satellite observations. We change the y-axis to "Stacked fraction of combustion AOD".

11. Figure 2: Shouldn't the units here for the y axes be just Tg?

The units must be specified with sulfur (S) or carbon (C).

12. Figure 6: Add the trend lines and trends for each plot.

The trends apparently depend on time period of the data record. Adding the trend lines makes the plots too heavy.

13. Figure 8: Why no CAM5 for 2016-2017? Is the MODIS trend only up to 2015, or does it include 2016-2017? It is of concern if the trends are computed for different time periods. This same comment can be applied to some of the later figures as well (see below).

Our CAM5 simulations stopped in 2015 because 2016-2017 emissions are not available. Data points for individual years are clearly shown in figures so that readers can see how 2016 and 2017 data are trending compared to previous years. Although the MODIS trend statistics include 2016 and 2017, this inclusion would not significantly affect our conclusions. Note also that significance tests of trend were performed by using

[Figure]

actual number of samplings (e.g., 15 and 13 for MODIS and CAM5, respectively).

14. Figures 9, 10, and 11: same concerns/questions/comments as Figure 8.

See response to #13.

15. Figure 11: How come no climatological seasonal cycle plots are shown, as in Figs. 8-10?

In this revision, we have added the climatological seasonal cycle plots for these regions affected by biomass burning smoke. Because of this addition, we have split the figure into Figure 11 (GOG and SAT) and Figure 12 (SEA). Appropriate discussion about seasonal cycles has been added.

16. Figure 12: Are you comparing trends from different time periods?

MODIS data include two extra years (2016 and 2017). Given that the individual data points are shown in figures, agreement and disagreement between the two datasets are evident.

Technical corrections: 1. Page 4, Line 120: "sea-spay" should be "sea-spray". Fixed.

2. Page 9, Line 268: The "0.025" here should be "0.015", correct? It should be 0.025.

3. Page 11, Line 340: I would switch "well captures" to "captures well". Fixed.

4. Page 11, Line 343: "the peak occurs" : : : the peak what? Be specific. Change to "the peak of combustion AOD occurs . . .."

5. Page 11, Line 343: Remove "the" before "MODIS" and "CAM5". Fixed.

6. Page 11, Line 345: Add "the" before "NAT", "MED", and "NAT and MED regions". Fixed.

7. Page 11, Line 356: "season" should be "seasons". Fixed.

8. Page 12, Line 369: Remove "the" before "CALIOP". Fixed.

9. Page 12, Line 370: Remove "the" before "SeaWiFS". Fixed.

10. Page 13, Line 405: Remove "the" before "CALIOP". Fixed.

11. Page 13, Lines 406-407: Define PM2.5, SO2, and CO. Fixed.

12. Page 13, Line 407: "decline" should be "declining". Fixed.

13. Page 13, Line 408: "decline" should be "declining". Fixed.

14. Page 13, Line 409: I would reword this sentence by changing "How" to "The manner in which". Fixed.

15. Page 13, Line 410: Why is "Figure 10" in bold print? Fixed.

16. Page 13, Line 417: "possibly" should be "possible". Fixed.

17. Page 13, Line 422: Remove "the" before "MODIS". Fixed.

18. Page 13, Line 426: Define ATSR. "Along-Track Scanning Radiometer". Defined.

19. Page 14, Line 459: Remove "the" before "northern". Fixed.

20. Page 14, Line 460: Remove "the" before "northern". Fixed.

21. Page 15, Line 461: Remove "the" before "southern". Fixed.

22. Page 15, Line 462: Remove "the" before "northern". Fixed.

23. Page 15, Line 468: In this context, I think "driving up" should be "drove up", although I suggest finding another phrase. Fixed.

24. Page 15, Line 490: Add "the" before "literature". Fixed.

25. Page 16, Line 496: Remove "the" before "CALIOP". Fixed.

26. Page 16, Line 500: Add "of" after "factor". Fixed.

27. Page 16, Line 506: Collection 6.1, correct? We used Collection 6, not 6.1. As we

discussed earlier, it will not affect major conclusions of this study.

28. Page 16, Line 507: Replace "the" after "over" with "a". Fixed.

29. Page 17, Line 528: Remove "only". Fixed.

30. Page 18, Line 566: Add commas after "Asia" and "China". Fixed.

31. Page 18, Line 570: Add "the" after "of". Fixed.

32. Page 18, Line 585: Replace "the" after "such" with "a". Fixed.

33. Page 26, Table 3 caption: Add "an" before "asterisk". Fixed.

34. Page 26, Table 4 caption: In the first line, "trend" should be "trends". Add a comma before "respectively". Also, add "an" before "asterisk". Fixed.

35. Page 27, Figure 1: (a) Add a label to the color bar. (b) Add latitudes and longitudes to the map. Added.

36. Page 28, Figure 2: Add labels (a)-(c) to the plots and the figure caption. The figure is labeled with (a), (b), and (c).

37. Page 29, Figure 3: I suggest using (a)-(c) for labels. Please also edit the caption accordingly. Also, add the difference between the marine cases to the caption. The figure is labeled and the caption is modified accordingly. The marine case is categorized into marine1 and marine2, representing different season.

38. Page 30, Figure 4: Add labels (a)-(h) to the plots and the figure caption. The figure is labeled and the caption is changed accordingly.

39. Page 32, Figure 6: Add labels (a)-(d) to the plots and the figure caption. The figure is labeled and the caption is changed accordingly.

40. Page 34, Figure 8: Add labels (a)-(f) to the plots and the figure caption. Also, add labels to the y axes of the middle plots. Add "by" after "simulated" in the last line of the figure caption. The figure is replotted with y-axis labels and the caption is revised

accordingly.

41. Page 35, Figure 9: Add labels (a)-(f) to the plots and the figure caption. Also, add labels to the y axes of the middle plots. The figure is replotted with y-axis labels and the caption is revised accordingly.

42. Page 36, Figure 10: Add labels (a)-(f) to the plots and the figure caption. Also, add labels to the y axes of the middle plots. The figure is replotted with y-axis labels and the caption is revised accordingly.

43. Page 37, Figure 11: Add labels (a)-(f) to the plots and the figure caption. Also, add labels to y axes of the plots on the left-hand-side of the figure. The figure is replotted with y-axis labels and the caption is revised accordingly.

44. Page 38, Figure 12: Label the x and y axes for each plot. Also, add labels (a)-(f) to the plots and the figure caption. In the figure caption, remove "and" before "the tropical Indian Ocean". The figure is replotted with y-axis labels and the caption is revised accordingly. The "and" before "the tropical Indian Ocean" is removed.

---

## Author Comment (AC3) · 8 Nov 2019

Referee #3 The authors present an evaluation of model and satellite observations of mineral dust/combustion aerosols over a fifteen -year period. They discuss the strengths/weaknesses of the MODIS products and various versions of the CAM simulations. In this manuscript further highlight the need to consistently use MODIS data to avoid errors, and the MODIS/Terra data are not to be used for AOD trends. Further, the conclusions regarding dust model simulations needing significant improvement are encouraging.

[Figure]

Overall, the manuscript is a challenging read due to the flamboyant reference of statistics. It is understood that the authors are putting the current evaluation in context of previous work, but it significantly impacts the readability of the manuscript. Other than the manuscript's readability, my comments on the paper are minor and I believe it should be published after minor revisions.

We thank the reviewer for comments. We will strive to streamline the text to better the flow of paper.

– Page 6, line 202: Can the authors clarify whether these differences are statistically significant or not?

These differences are based on an analysis of a large amount of data points and are substantially larger than standard errors. The bottom line here is that we should use platform-specific numbers to attain the self-consistent use of the data.

– After reading the manuscript I could not conclude which (MODIS or CAM5) is more reliable. Page 12, lines 393-398 and section 3.3.3 left me somewhat confused. Can the authors clarify which is a "better" product according to their research?

Each product has its own strength and weakness, which depends on region. In our analysis, we have been trying to discuss major uncertainties associated with individual product and assess which product is better based on the discussion and independent data if available. For example, MODIS-detected dust decreasing trend in Northwestern Pacific Ocean (NWP) is consistent with independent measurements over Asian dust source regions and Japan (AD-Net lidars). But CAM5 model doesn't capture this trend and we believe that the model needs improvement of Asian dust. Another example is the Southeast Asia (SEA) outflow region. CAM5 POM emission and combustion AOD were significantly smaller than 2014, which is contradict with well-documented intense wildfires in 2015. We thus believe that the model's fire emissions are likely underestimated. Unfortunately, it is not always possible to judge which product is more reliable due to lack of independent data sets to collaborate the MODIS or CAM5 results.

– Page 17, line 561: I am confused regarding the use of "monotonous" here. Please consider changing.

"monotonous" is removed.

– Page 18: line 583: "...processes, such the broad..." should be "...processes, such a broad..."?

Fixed

– Figure 6 caption: Capitalize first word in sentence.

Fixed.